# Profiling Pareto Front With Multi-Objective Stein Variational Gradient Descent

**Xingchao Liu** [*]
Department of Computer Science
University of Texas at Austin
xcliu@utexas.edu

**Xin T. Tong**
Department of Mathematics
National University of Singapore
mattxin@nus.edu.sg

**Qiang Liu**
Department of Computer Science
University of Texas at Austin
lqiang@cs.texas.edu

## Abstract

Finding diverse and representative Pareto solutions from the Pareto front is a key challenge in multi-objective optimization (MOO). In this work, we propose a novel gradient-based algorithm for profiling Pareto front by using Stein variational gradient descent (SVGD). We also provide a counterpart of our method based on Langevin dynamics. Our methods iteratively update a set of points in a parallel fashion to push them towards the Pareto front using multiple gradient descent, while encouraging the diversity between the particles by using the repulsive force mechanism in SVGD, or diffusion noise in Langevin dynamics. Compared with existing gradient-based methods that require predefined preference functions, our method can work efficiently in high dimensional problems, and can obtain more diverse solutions evenly distributed in the Pareto front. Moreover, our methods are theoretically guaranteed to converge to the Pareto front. We demonstrate the effectiveness of our method, especially the SVGD algorithm, through extensive experiments, showing its superiority over existing gradient-based algorithms.

## 1  Introduction

Many scientific and engineering problems involve optimizing multiple conflicting objectives [5, 28, 4, 24], including, for example, designing wireless sensors [12], building electric power systems [30], and training neural networks with multiple tasks [35]. With multiple conflicting objectives, it is impossible to find a single solution that optimizes all the objectives simultaneously. Instead, it is essential to find a set of diverse solutions in the Pareto front that represent different preferences on the different objective functions, so that the users can have a global view of how the optimal trade-off of the different objectives look like and select the solution according to their own preference.

Unfortunately, profiling the Pareto fronts casts a key computational challenge, especially for high dimensional problems [5, 12, 28]. Traditionally, a large literature has been devoted to developing black-box, derivative-free algorithms that are suitable for black-box optimization, such as these based on evolutionary algorithms [8] and Bayesian optimization [25, 1, 39]. However, the black-box algorithms tend to be expensive and can only be applied to small scale problems due to the lack of gradient information. Gradient-based MOO algorithms have been catching attention only recently, which include mainly multiple gradient descent (MGD) based methods [10, 22, 27]. However,

---

[*]Code is available at https://github.com/gnobitab/MultiObjectiveSampling.

35th Conference on Neural Information Processing Systems (NeurIPS 2021).

the results of these methods are still dissatisfying in finding diverse and evenly distributed Pareto solutions in complex problems.

We introduce Stein variational gradient descent (SVGD) [20, 19, 21] and Langevin dynamics [37] as efficient approaches for profiling Pareto fronts. These methods iteratively evolve a group of particles to represent the target distribution. The main difference of the two algorithms is how they distribute points. The Langevin dynamics uses stochastic noises to perturb the particle trajectories, so they can visit different areas of the Pareto front. On contrast, SVGD is a deterministic sampling algorithm that pushes the particles to high probability regions using gradient information, while enforcing diversity between the particles using a repulsive force.

In this work, we propose to a simple approach to integrate SVGD and Langevin Dynamics with MGD to draw samples from the Pareto front. Theoretical analyses are provided for both algorithms to understand their limiting distributions and their convergence speed. One challenge with MGD based sampling is that the limiting distributions do not admit explicit formulations. This is mainly because the forcing from MGD is in general not the gradient of any function. However, we can show its non-gradient component is orthogonal to a large class of functions. Assuming each objective function is strongly convex and regular, we can also show the limiting distributions concentrate on the Pareto front. Moreover, we can show the two algorithms converge to good solutions of MOO with $O(1/t)$ and linear rate.

We test our methods on a variety of tasks, ranging from low-dimensional optimization to multi-task neural network optimization. On all the tasks tested, our method can obtain diverse and high quality Pareto solutions that distribute evenly on the Pareto front, without predefined preference vectors. Quantitatively, we substantially outperform PF-SMG and EPO with respect to the hypervolume metric.

## 2  Background

In Multi-objective optimization (MOO), we are interested in minimizing a vector-valued loss function $F(x) = (f_1(x), f_2(x), \ldots, f_m(x)) \in \mathbb{R}^m$, where $x \in \mathbb{R}^d$ and $f_i : \mathbb{R}^d \to \mathbb{R}, \ i \in [m]$ is the $i$-th scalar-valued objective function. Here we use notation $[m] = \{1, 2, \ldots, m\}$. Obviously, we can not fully optimize all the objective functions simultaneously because they may be conflicting with each other. Instead, we are interested in finding the points which can not be improved simultaneously for the objective functions, yielding the notion of *Pareto optimality*.

**Definition 1** ( Pareto Optimality)**.** *For $x_1, x_2 \in \mathbb{R}^d$, We say that $x_1$ is* dominated *by $x_2$ iff $f_i(x_2) \leq f_i(x_1), \forall i \in [m]$, and $F(x_1) \neq F(x_2)$. A point $x^*$ is called* globally Pareto optimal *on $\mathbb{R}^d$ iff it is not dominated by any other $x' \in \mathbb{R}^d$. A point $x^*$ is called* locally Pareto optimal *iff there exists an open neighborhood $\mathcal{N}(x^*)$ of $x^*$, such that $x^*$ is not dominated by any $x \in \mathcal{N}(x^*)$. The collection of globally (resp. locally) Pareto optimal points are called the global (resp. local) Pareto set. The collection of function values $F(x^*)$ of all the Pareto points $x^*$ is called the Pareto front.*

Our goal is to find diverse and representative solutions $\{x_i\}_{i=1}^n$ from the Pareto set, so that their function values $\{F(x_i)\}_{i=1}^n$ covers different preferences on different objectives. This would allow the end-users to have a global view on the optimal trade-off between the different objective functions and decide which solution based on their own preference.

**Linear Scalarization**   One standard approach to solve MOO is using a preference vector $\lambda = [\lambda_1, \ldots, \lambda_m]$ from the probability simplex on $[m]$, i.e., $\mathcal{S} = \{\lambda : \sum_{i=1}^m \lambda_i = 1, \lambda_i \geq 0, i \in [m]\}$. Each $\lambda \in \mathcal{S}$ leads to a weighted objective function $f_\lambda(x) = \sum_{i=1}^m \lambda_i f_i(x)$ and its minimizer $x_\lambda^* = \arg\min_x f_\lambda(x)$. Then as we take $\lambda$ in a grid of $\mathcal{S}$, we hope that the corresponding $x_\lambda^*$ gives a rough sketching of the Pareto front.

Although this strategy is simple and easy to implemented, it suffers from the a number of weaknesses. A key problem is that $F(x_\lambda^*)$ can only lie on the convex envelope of the Pareto front, and hence it only works in cases when the Pareto front is convex. In addition, a uniform grid of $\lambda$ on $\mathcal{S}$ does not necessarily yield uniformly distributed points on the Pareto front.

**Multiple Gradient Descent (MGD)**   MGD is a natural extension of the single-objective gradient descent to finding a Pareto point [10], which, unlike linear scalarization, works for non-convex Pareto

fronts. The idea is to iteratively update the variable $x$ along a direction that ensures that *all the objectives are decreased simultaneously* (which is called Pareto improvement).

Let $g_i(x) = \nabla f_i(x)$ be the gradient of the $i$-th objective. Suppose we update the variable by $x' \leftarrow x - \epsilon g^*(x)$, where $g^*(x)$ is a vector field to be determined and $\epsilon$ is a small step size. By Taylor approximation, we have $\langle g_i, \; g^* \rangle \approx -(f_i(x') - f_i(x))/\epsilon$, which represents the decreasing rate of $f_i$ when we update $x$ along direction $g^*(x)$. In MGD, $g^*$ is chosen to maximize the slowest decreasing rate among all the objectives, that is,

$$g^*(x) \propto \arg\max_{g \in \mathbb{R}^d} \left\{ \min_{i \in [m]} \langle g, g_i(x) \rangle, \;\; s.t. \;\; \|g\| \leq 1 \right\}. \tag{1}$$

Therefore, $g^*(x)$ is encouraged to have positive inner products with all $g_i(x)$. If this is impossible to achieve, $\{g_i(x)\}_{i=1}^m$ will contain conflicting directions, and we would have $g^*(x) = 0$ which terminates the algorithm. Using Lagrangian duality, we can show that the optimal solution to (1) is $g^*(x) \propto \sum_{i=1}^m \lambda_i^*(x) g_i(x)$, where $\{\lambda_i^*(x)\}_{i=1}^m$ is the solution of

$$\min_{\{\lambda_i\}} \left\| \sum_{i=1}^m \lambda_i g_i(x) \right\| \; s.t. \; \sum_{i=1}^m \lambda_i = 1, \; \lambda_i \geq 0, \; \forall i. \tag{2}$$

This optimization has a simple closed-form solution when $m = 2$, and a fast algorithm is offered by [35] for $m > 2$. By construction, when the step size $\epsilon$ is small, MGD monotonically decreases *all* the objectives simultaneously and will terminate when it arrives a local Pareto point; in this case we have $g^*(x) = \sum_{i=1}^m \lambda_i^*(x) g_i(x) = 0$, suggesting that the zero vector 0 is inside the convex hull of $\{g_i(x)\}_{i=1}^m$.

The MGD direction $g^*$ provides a natural notion of gradient for multiple objectives. A point $x$ is said to be Pareto stationary if $g^*(x) = 0$. Similar to the case of differentiable single-objective optimization, every Pareto local optimal is a Pareto stationary point. See [10, 35, 22] for more discussion.

However, the vanilla MGD suffers from several key weaknesses. Although it promises to return a point on the (local) Pareto set, and it is difficult to explicitly control which Pareto point it will converge to. The convergence point of MGD is implicitly determined by the initialization and the other hyper-parameters of the algorithm (e.g., step size) in a complicated way. It can be harder for MGD to converge to some Pareto points than the others. In fact, assume the Pareto front is an open set in $\mathbb{R}^d$, then the vanilla MGD, when initialized from outside of Pareto front, will terminate when it reaches the boundary of the Pareto front, and hence *never reach the interior points*. Below is a simple example to demonstrate this.

**Example 1.** *Consider $f_1(x) = x^2$ and $f_2(x) = (x-1)^2$ for $x \in \mathbb{R}$ as shown in Figure 1. Then the Pareto set is the interval $[0, 1]$. However, when initialized outside of $[0, 1]$, MGD will converge to either $x = 0$ or $x = 1$ and can not reach the interior points of the Pareto set unless we add a diversity-promoting mechanism, such as random noise or deterministic repulsive force. See figure 1.*

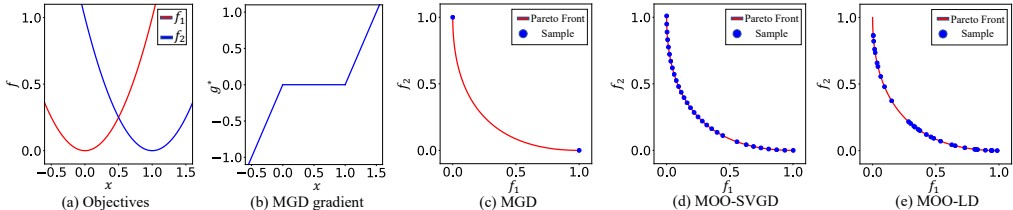

(a) Objectives      (b) MGD gradient      (c) MGD      (d) MOO-SVGD      (e) MOO-LD

Figure 1: (a) The plot of $f_1$ and $f_2$ in Example 1. (b) The MGD direction $g^*(x)$. The Pareto set is $[0, 1]$, within which we have $g^*(x) = 0$. (c) MGD can only converge to either of the two end points ($x = 0$ or $x = 1$) of the Pareto set $[0, 1]$ when initialized outside of $[0, 1]$. (d) Solutions from our MOO-SVGD; we use 30 particles with 3000 iterations. (e) Solutions from MOO-LD; we run the algorithm for $10^6$ steps, and uniformly sample 30 points from the last 1000 points.

## 3 Profiling Pareto front with Particle Dynamics

We improve MGD by integrating it with sampling algorithms for approximating distributions to obtain diverse solutions that cover the Pareto front more evenly. In this section, we first introduce our

main algorithm, which integrates MGD with Stein variational gradient descent (SVGD), yielding an interacting particle system that evolves diverse Pareto solutions with deterministic repulsive forces (Section 3.1). We also introduce a related idea of combining MGD with Langevin dynamics to obtain diverse solutions by injecting diffusion noises, which is found less favorable in practice than SVGD-based method but is of independent interest (Section 3.2).

## 3.1 Multi-Objective Stein Variational Gradient Descent

Stein Variational Gradient Descent (SVGD) [20] is a deterministic sampling method for finding diverse and representative points (a.k.a. particles) for approximating an un-normalized distribution, by iteratively evolving the particles with both gradient information and a special repulsive force. In this section, we integrate SVGD with MGD and adapt it to profile the Pareto front.

We derive our algorithm following how SVGD was derived in [20]. We initialize a set of points $\{x_i\}_{i=1}^n$ (a.k.a. particles) and iteratively updates them to 1) mimic the MGD trajectory and 2) maintain diversity between the points. For the derivation, assume the particles $\{x_i\}$ follow a distribution $\rho$ and we update them with $x_i \leftarrow x_i - \epsilon \phi^*(x)$, $\forall i \in [n]$, where $\phi$ is an update direction, selected from a predefined function space $\mathcal{H}$, to strike the balance of the following two factors:

1) We want to choose $\phi$ to make the dynamics mimic the MGD trajectory as close as possible; this can be framed as maximizing the average inner product $\mathbb{E}_{x \sim \rho}[\langle g^*(x), \phi(x)\rangle]$, to make the direction of $\phi$ and $g^*$ as close as possible.

2) We also want to choose $\phi$ to encourage the diversity between the particles. Assume $\rho$ is a continuous distribution; the diversity of $\rho$ can be measured by the entropy $H(\rho)$, and hence we want to choose $\phi$ such that the entropy of the distribution of the updated particles, denoted by $\rho'$, is maximized. [20] showed that $H(\rho') = H(\rho) - \epsilon \mathbb{E}_{x \sim \rho}[\nabla \cdot \phi(x)] + O(\epsilon^2)$, where $\nabla \cdot$ denotes the divergence operator in vector calculus. Therefore, we should choose $\phi$ to maximize $-\mathbb{E}_{x \sim \rho}[\nabla \cdot \phi(x)]$ to encourage the diversity of the updated particles.

Overall, we optimize $\phi$ to maximize a linear combination of the two terms above: $\phi_\rho^* = \arg\max_{\phi \in \mathcal{F}} \{\mathbb{E}_{x \sim \rho}[\langle g^*(x), \phi(x)\rangle - \alpha \nabla \cdot \phi(x)]\}$, where $\alpha$ is a positive coefficient that controls the importance of the divergence term. Following [20], we take $\mathcal{F}$ to be the unit ball of a reproducing kernel Hilbert space (RKHS) with a positive definite kernel $k(x, x')$. With the same derivation as [20], we get a closed form solution: $\phi_\rho^*(x) \propto \mathbb{E}_{x' \sim \rho}[g^*(x')k(x', x) - \alpha \nabla_{x'} k(x', x)]$.

By approximating $\rho$ with the empirical distribution of the particles $\rho = \sum_i \delta_{x_i}/n$ and iteratively updating the particles with $\phi_\rho^*$ above, we obtain the following MOO-SVGD:

$$x_i \leftarrow x_i - \epsilon \hat{\phi}(x_i), \quad \text{where} \quad \hat{\phi}(x_i) = \frac{1}{n}\sum_{j=1}^n g^*(x_j)k(x_i, x_j) - \alpha \nabla_{x_j} k(x_i, x_j). \tag{3}$$

We can see that MOO-SVGD simply replaces the gradient in SVGD with the multiple gradient $g^*$.

The intuition of the update is clear: the first term in $\hat{\phi}$ makes the dynamic follow the MGD direction so that the particles are pushed towards the Pareto set when $\{g^*(x_i)\}$ have large magnitudes; the second term pushes the different particles away from each other, so that more areas of the Pareto set is approximated. In particular, when all the particles are in the Pareto set with $g^*(x_i) \approx 0$, then only the pairwise repulsive force will push the particles to be away from each other and form a uniform distribution inside the Pareto set.

If $g^*$ is the gradient vector of some scalar function, i.e., $g^*(x) = \nabla f^*(x)$, then our method reduces to the standard SVGD for sampling from distribution $\rho^*(x) \propto \exp(-f^*(x)/\alpha)$. In general, $g^*$ is not a gradient vector field of any scalar-valued objective function, and hence our method yields a general "non-gradient" variant SVGD. Similar to vanilla SVGD, we can characterize the evolution of the density of the particles with a nonlinear different equation. In the large particle ($n \to \infty$) and continuous time limit, let $\rho_t$ be the limit density of the particles at time $t$ ($\epsilon \to 0$). Following [19], the evolution of $\rho_t$ is governed by,

$$\frac{d\rho_t(x)}{dt} = \nabla_x \cdot (\phi_\rho^*(x)\rho_t(x)). \tag{4}$$

Unlike the case of the vanilla SVGD, the behavior of (4) is less clear due to the non-gradient nature of $g^*(x)$. In the following, we show that the stationary distribution of (4) is connected to a Helmholtz

like decomposition of $g^*(x)$ into a gradient component and non-gradient components. To facilitate our discussion, we define the kernel embedding of any function $f$ with density $\rho$ as

$$f_{[\rho]}(x) = \int k(x, y) f(y) \rho(y) dy.$$

**Theorem 3.1.** *Assume $\rho^*$ is a fixed point of* (4) *for which $\log \rho^*(x)$ is continously differentiable and satisfies the Stein's identity $\mathbb{E}_{x \sim \rho^*}[\nabla \log \rho^*(x) k(x, x') + \nabla k(x, x')] = 0$ for every $x' \in \mathbb{R}^d$. Then we have the following decomposition of the vector field $g^*$:*

$$g^*(x) = -\alpha \nabla \log \rho^*(x) + \jmath(x), \tag{5}$$

*where $\nabla \log \rho^*$ and $\jmath$ can be viewed as the gradient and non-gradient components of $g^*$, and $\jmath(x)$ satisfies the following property: its kernel embedding is orthogonal to the kernel embedding of any gradient field in RKHS $\mathcal{H}$, that is, let $h$ be any $C^1$ function with a compact support then $\langle \jmath_{[\rho^*]}, \nabla h_{[\rho^*]} \rangle_{\mathcal{H}} = 0$.*

The existence of $\rho^*$ is discussed in the Appendix after the proof of Theorem 3.1. Next, we show that $\rho^*$ concentrates on the Pareto set. In order to do so, recall that $\|g^*(x)\| = 0$ indicates that $x$ is on the Pareto set. Therefore we will show the norm of the MGD field $g^*$ is small over $\rho^*$ and SVGD updates can find a solution with small $\|g^*\|$.

To simplify our discussion, we limit our discussion on the case where a truncated $(\sigma, m_d)$-Gaussian kernel is used: $k(x, y) = (\det(2\pi\sigma^2 I))^{-\frac{1}{2}} \exp(-\frac{1}{2\sigma^2} \|x - y\|^2) 1_{\|x-y\| \leq m_d \sigma}$. Note that its gradient may be singular at the truncation threshold. Formally we can ignore such singularity and simply define formally: $\nabla_y k(x, y) := \frac{x-y}{\sigma^2} k(x, y) 1_{\|x-y\| \leq m_d \sigma}$. We also assume the objectives functions are regular and strongly convex:

**Assumption 1.** *Each objective function $f_i$ is $C^3$, i.e., third-order continuously differentiable, and $c_1 I \preceq \nabla^2 f_i \preceq c_2 I$, $|\partial^3_{j,k,l} f_i| \leq c_3$, for $i, k, l \in [d]$, where $0 < c_1 \leq c_2 \leq \infty$, $0 \leq c_3 < \infty$ are constants. Here $\nabla^2 f_i$ is the Hessian matrix of $f_i$, and $A \preceq B$ indicates that $B - A$ is a positive semi-definite matrix.*

It is well known that, with strongly convex single objective function, gradient based algorithm can converge to the global minimum. We show that this also true for MOO-SVGD method.

**Theorem 3.2.** *Suppose $\|g^*\|^2$ is $C^1$, $g^*$ is $L$-Lipschitz and Assumption 1 holds. If we apply MOO-SVGD using the $(\sigma, m_d)$-truncated Gaussian kernel $k$, the equilibrium distribution $\rho^*$ of MOO-SVGD satisfies*

$$\|g^*_{[\rho^*]}\|^2_{\mathcal{H}} = \int \rho^*(x) k(x, y) \rho^*(y) g^*(x)^\top g^*(y) dx dy$$

$$\leq \int \rho^*(x) k(x, y) \rho^*(y) \|g^*(x)\|^2 dx dy \leq \frac{1}{2c_1^2} (L + \alpha/\sigma^2)^2 m_d^2 \sigma^2 \|1_{[\rho^*]}\|^2_{\mathcal{H}}.$$

*If the MOO-SVGD density $\rho_t$ following* (4) *is considered, for any $t$ there is an $s \leq t$ such that*

$$\|g^*_{[\rho_s]}(x)\|^2_{\mathcal{H}} \leq \frac{1}{2c_1^2} (L + \alpha/\sigma^2)^2 m_d^2 \sigma^2 \|1_{[\rho_s]}\|^2_{\mathcal{H}} + \frac{1}{t} \mathbb{E}_{x \sim \rho_0} \|g^*(x)\|^2.$$

Claim 1 of Theorem 3.2 says that, at the equilibrium $\rho^*$, the kernel embedding of $g^*$ is $O((1 + \alpha/\sigma^2)^2 \sigma^2)$ which can be arbitrarily small if $\sigma$ and $\alpha$ are chosen to be small numbers. Claim 2 of Theorem 3.2 indicates the SVGD density flow $\rho_t$ can find a solution of similar property in finite time.

## 3.2 Multi-Objective Langevin Dynamics

Besides SVGD, MGD can also be integrated with Langevin dynamics (LD), which yields diverse solutions by injecting random noise into the updates. Similar to SVGD, we can achieve this by simply replacing the gradient field in LD with $g^*$, yielding MOO-LD:

$$x \leftarrow x - \epsilon g^*(x) + \sqrt{2\alpha\epsilon}\xi, \tag{6}$$

where $\xi$ is a standard Gaussian noise and $\alpha$ is a positive coefficient that determines the magnitude of the noise. The intuition is again simple: if $x$ is far away from the Pareto set with a large multiple gradient $\|g^*(x)\|$, the dynamics will mainly drive $x$ towards the Pareto set; when $x$ is close to Pareto set with a small $\|g^*(x)\|$, the noise term will dominate and allow $x$ randomly diffuse on Pareto set.

If $g^*$ is the gradient vector of some sclar-valued function $f^*$, that is, $g^*(x) = \nabla f^*(x)$, then its stationary distribution equals $\rho^*(x) \propto \exp(-f^*(x)/\alpha)$. If $g^*$ is not a gradient field, the Langevin dynamics is known to be irreversible [e.g., 32, 11].

As shown in our empirical results, the SVGD based method is more practically attractive than the Langevin method. We find that SVGD tends to provide more diverse and uniformly distributed points than Langevin dynamics. And for some of the challenge MOO benchmarks, Langevin dynamics can be stuck in a small local region and is slow to mix. In comparison, thanks to the deterministic updates, SVGD is found to generate mores uniformly distributed Pareto solutions in a faster speed. Another practical advantage of SVGD is that we can define the kernel to be of form $k(x, y) = k_0(F(x), F(y))$, so that the diversity force is w.r.t. the function values $\{F(x_i)\}$, rather than the variables $\{x_i\}$.

In the following, we consider the asymptotic property of the dynamics in (6). Consider the continuous time limit of the dynamics in (6) ($\epsilon \to 0$), which yields a stochastic differential equation

$$dx_t = -g^*(x_t)dt + \sqrt{2\alpha}dw_t, \tag{7}$$

where $w_t$ denotes standard Wiener process and $t$ denotes time. Assume we initialize $x_0$ from a distribution with continuous differential density function $\rho_0$, then the evolution the density of $x(t)$, denoted by $\rho_t$, is governed by a Fokker-Planck equation:

$$\frac{d}{dt}\rho_t(x) = \nabla \cdot (g^*(x)\rho_t(x) + \alpha\nabla\rho_t(x)), \tag{8}$$

where $\nabla\cdot$ denotes the divergence operator in vector calculus.

The stationary distribution of (7) and (8), denoted as $\rho^*$, uniquely exists as long as $g^*(x)$ is coercive (e.g., if Assumption 1 holds). See e.g., [29]. It yields a decomposition of $g^*(x)$ analogous to that of (5). Moreover, we can show the KL divergence between $\rho_t$ and $\rho^*$ decays:

**Theorem 3.3.** *Assume $\rho^*$ is a fixed point of (8) for which $\log \rho^*(x)$ is in $C^1(\mathbb{R}^d)$. Then we have the following Helmholtz decomposition of the vector field $g^*$: $g^*(x) = -\alpha\nabla \log \rho^*(x) + \jmath(x)$, where $\jmath$ satisfies that, for any $C^1(\mathbb{R}^d)$ function $h$ with a compact support, $\jmath$ is orthogonal with $\nabla h$ under $\rho^*$: $\int \jmath(x)^\top \nabla h(x)\rho^*(x)dx = 0$. Further, if $\rho_t(x)\jmath(x) \to 0$ when $\|x\| \to \infty$, the KL divergence between $\rho_t$ and $\rho^*$ decreases monotonically with*

$$\frac{d}{dt}\mathrm{KL}(\rho_t \,||\, \rho^*) = -\alpha\mathrm{F}(\rho_t \,||\, \rho^*), \tag{9}$$

*where $\mathrm{F}(\rho_t \,||\, \rho^*) = \mathbb{E}_{x\sim\rho_t}[\|\nabla \log(\rho_t(x)/\rho^*(x))\|^2]$ is the Fisher divergence.*

Theorem 3.3 is similar to Theorem 3.1. The differences are: 1) there is no kernel, and the decomposition is more well-understood, see for example Section 1 of [11]; 2) Interestingly, SVGD does not seem to have a similar monotonic decreasing property similar to (9). In particular, if $\rho^*$ is strongly log-concave, it is well known that (9) indicates that $\rho_t$ converges to $\rho^*$ linearly in KL divergence.

Analogues to Theorem 3.2, we can also show the MOO-LD yields a distribution that concentrates on the Pareto (stationary) set. In fact, we can further show $\|g^*(x)\|$ is sub-Gaussian under $\rho^*$.

**Theorem 3.4.** *Under Assumption 1, let $M = c_3^2\alpha/c_1 + c_2^2$. For any constant $0 < b \leq \frac{c_1}{4\alpha c_2^2}$, the fixed point $\rho^*$ to (8) satisfies*

$$\mathbb{E}_{x\sim\rho^*}\|g^*(x)\|^2 \leq 4\alpha Md/c_1, \quad \mathbb{E}_{x\sim\rho^*}\exp(b\|g^*(x)\|^2) \leq 3\exp(16\alpha bMd/c_1).$$

*The solution $\rho_t$ to (8) satisfies the following*

$$\mathbb{E}_{x\sim\rho_t}\|g^*(x)\|^2 \leq \exp(-\frac{1}{2}c_1 t)\mathbb{E}_{x\sim\rho_0}\|g^*(x_0)\|^2 + 4\alpha Md/c_1,$$

$$\mathbb{E}_{x\sim\rho_t}\exp(b\|g^*(x)\|^2) \leq \exp(-\alpha bMdt)\mathbb{E}_{x\sim\rho_0}\exp(b\|g^*(x)\|^2) + 3\exp(16\alpha bMd/c_1).$$

# 4 Related Works

We discuss related works on MOO and also introduce the hypervolume (HV) indicator which we will use as the evaluation metric in our experiments.

**Gradient-free MOO Algorithms**  A large volume of literature focuses on black box and derivative-free MOO, where $f_i(x), \ i \in \{1, 2, \ldots, m\}$ are observed through the outputs [e.g., 8, 25, 15, 39, 25, 13]. These methods are typically based on evolutionary algorithm or Bayesian optimization. However, due to the lack of gradient information, they can perform well only in low dimensional problems and can not be applied to large-scale optimization problems in modern machine learning, such as learning neural networks.

**Gradient-based MOO Algorithms**  Another important line of work is gradient-based methods. The foundation of these methods is MGD. Representative methods include Pareto Front Stochastic Multiple Gradient (PF-SMG) [10, 22], Pareto Multi-task Learning (PMTL) [18], and Exact Pareto Optimization (EPO) [27].

PF-SMG leverages the randomness in stochastic multiple gradient descent to profile the whole Pareto front. The method keeps a set of solutions. In every optimization step, the algorithm removes dominated solutions from the set, randomly sample new solutions from the neighborhood of the survived solutions, and run MGD for another multiple steps. PF-SMG is similar to the Langevin method we discuss in Section 3.2 in that both of them leverage random noise, but PF-SMG leverages the noise in stochastic gradient, which will vanish to zero when the step size decreases to zero. Therefore, PF-SMG would reduce to MGD except the selection step. In constrast, the diffusion noise converges to Brownian motion with small step size.

PMTL and EPO are designed to find local Pareto optimal solutions that satisfy specific user preference. They can also be used to trace the Pareto front by varying the user preference vector uniformly. However, sampling the preference vector uniformly will not necessarily result in a uniform solution set on the Pareto front, which is what we observe in the experiments. Compared with PTML and EPO, our method does not require pre-defined user preference and can obtain more diverse and uniformly distributed points as we show in experiments.

**Pareto Hypernet Methods**  Several methods, such as [31, 17, 33], proposed to learn a neural network, called Pareto hypernet, which can generate pareto-optimal solutions given a preference vector as input. In comparison, our methods directly generates a set of solutions (or particles) that covers the Pareto front. The usage of neural network can be helpful in some cases, but it also makes the algorithm more complicated and less transparent, and the choice of neural network structure has heavy influence on the result. In comparison, our algorithm is much simpler, transparent and directly outputs the solutions that we want. More empirical evidence and discussion can be found in the Appendix.

**Hypervolume**  The hypervolume (HV) indicator is a standard metric for evaluating the quality of sets of solutions in MOO. Assume $r = [r_1, \ldots, r_m]$ is an reference point that is an upper bound of the objectives, such that $\sup_x f_i(x) \leq r_i, \ \forall i \in [m]$. For a given set of solutions $\mathcal{X} = \{x_i\}_{i=1}^n$, its hypervolume indicator $HV(\mathcal{X})$ is the measure of the region between $F(\mathcal{X})$ and $r$,

$$HV(\mathcal{X}) = \Lambda \left( \left\{ q \in \mathbb{R}^d \mid \exists x \in \mathcal{X} : \ q \in \prod_{i=1}^m [f_i(x), r_i] \right\} \right), \tag{10}$$

where $\Lambda(\cdot)$ denotes the Lebesgue measure. Directly optimizing the hypervolume metric can be problematic, because it is piece-wise constant. Hypervolume indicator gradient ascent (HIGA) [36] relaxes the hypervolume indicator to get usable gradient for optimization, and a recent work [9] is mathematically similar to HIGA. However, [36] and [9] have two main drawbacks compared with our sampling-based method: (1) its results depend on the choice of reference point; (2) it may get stuck in bad local minima. See the Appendix for more empirical comparison.

# 5 Experiments

We show our method, Multi-Objective Stein Variational Gradient Descent (MOO-SVGD) can efficiently profile the Pareto front on a variety of problems, including the ZDT problem set, tri-objective problem MaF1, trade-off between accuracy and fairness metrics in fair ML, as well as multi-task

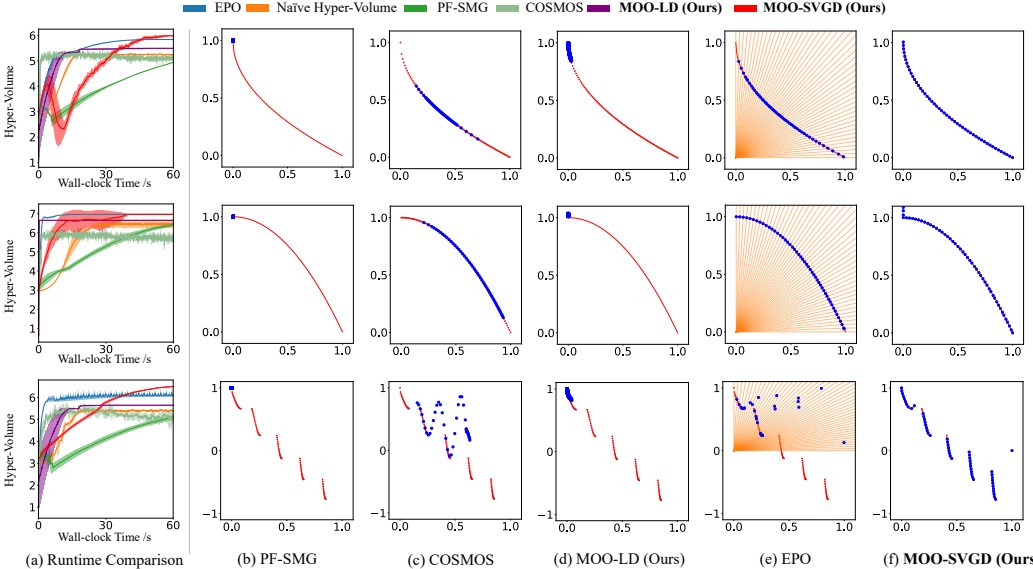

Figure 2: (a) The hypervolume of the results of different methods w.r.t. the wall clock time. (b) The solution set obtained by PF-SMG. All the points converge to the same solution because they do not interact with each other. (c) The solution set obtained by COSMOS. In ZDT1, uniform preference vector causes non-uniform distribution of the final solution; in ZDT3, the solutions are not on the real Pareto front because they are limited by the imperfect optimization of the function approximator. (d) The solution set obtained by MOO-LD. Thanks to the random noise, it has slightly larger coverage than PF-SMG, but it is still not widespread even with 200,000 samples. (e) The solution set obtained by EPO. The orange lines refer to the preference vectors. Note that in ZDT1, uniform preference vector causes non-uniform distribution of the final solution; in ZDT3, EPO cannot deal with negative objectives. (f) Benefited from the explicit repulsive force, MOO-SVGD obtains a uniform Pareto front. All the results are averaged over 5 runs.

learning on MultiMNIST, MultiFashion and MultiFashion+MNIST. Our method outperforms EPO and PF-SMG, two previous baselines for profiling the Pareto front with gradients. We also include Multi-Objective Langevin Dynamics (MOO-LD) for comparison, and find that MOO-SVGD yields much better empirical performance against MOO-LD. MOO-LD in general needs longer iterations to cover the Pareto Front. Please refer to the Appendix for more discussion. We leave the detailed algorithm configuration in the Appendix due to space limitation.

**ZDT Bi-objective Problems**  We compare our methods with other baseline methods on the ZDT problem set [40]. The ZDT problems have a 30-dimensional variable and two objective functions. The details of ZDT problems can be found in the Appendix. In ZDT problems, both objective functions are differentiable. We choose ZDT-1, ZDT-2, and ZDT-3 to demonstrate the performance of our algorithm when the Pareto front is convex, concave, and discontinuous. For baselines, we compare with naive optimization on hypervolume , state-of-the-art gradient-based MOO algorithms (EPO [27] PF-SMG [22]) and Pareto hypernet algorithms (COSMOS [33]). All the results of the baselines are reproduced using their official implementation. For fair comparison between black-box and white-box optimization algorithms, we report the increase of the hypervolume indicator (Eq. (10)) over the wall-clock time. For fairness of comparison, we use the same reference point when computing hypervolume indicator for every algorithm. Moreover, to make the comparison of wall-clock time fair, we run all the algorithms with CPU on the same computational platform with 192GB memory and 48 cores. The results are shown in Fig. 2. We constrain the maximal wall-clock time in the figure to be 60 seconds. We use 50 particles for MOO-SVGD. Because EPO and COSMOS need user preference vectors, we uniformly sample 50 unit vectors from a semi-circle to trace a Pareto front, following the strategy provided by the authors [27, 33]. We use a two-layer ReLU network with 1,000 hidden units for the hypernet in COSMOS.

As the figures shows, our method, MOO-SVGD, obtains the largest hyper-volume in all three problems. In the limit of 60 seconds, EPO generally has the best performance among all the three baselines. Naive optimization of hyper-volume get similar performance as the stochastic method, PF-SMG, though the latter is much slower than the former. COSMOS cannot cover the entire Pareto

front. In ZDT1 and ZDT3, MOO-SVGD converged to a better solution. In comparison, another sampling based method, MOO-LD, can only cover a limited subset of the whole Pareto front.

**MaF1 Tri-objective Problem** We test our algorithm on MaF1 [3], a problem with three objectives to showcase its ability when there are more than two objectives. The objective functions are defined on $[0, 1]^d$. The specific expression of $f_1$, $f_2$ and $f_3$ can be found in Appendix due to limited space. The Pareto front of the problem can be analytically solved, which is $\{x\colon f_1(x) + f_2(x) + f_3(x) = 2\}$. The reference point for calculating hypervolume is $(1.0, 1.0, 2.0)$. The results are shown in Fig 3 and Table. 1. All the three methods, EPO, MOO-LD, and MOO-SVGD can find Pareto optimal solutions. However, MOO-LD can only cover a very small subset of the Pareto front. EPO can find diverse Pareto optimal solutions, but they are not distributed uniformly. MOO-SVGD obtains the most visually uniform solution set, and hence reaches the highest performance w.r.t. hypervolume. MOO-LD is not tested in the following parts due to its poor performance on toy experiments.

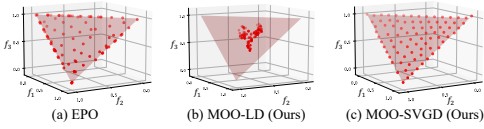

(a) EPO  (b) MOO-LD (Ours)  (c) MOO-SVGD (Ours)

Figure 3: The solution sets obtained by EPO, MOO-LD and MOO-SVGD. The red plane is the Pareto front. The red dots are the final solutions.

| Method | Hypervolume |
|---|---|
| EPO | 0.5714 |
| MOO-LD (Ours) | 0.2613 |
| **MOO-SVGD (Ours)** | **0.5867** |

Table 1: The Hyper-volume of the solution sets obtained by EPO, MOO-LD and MOO-SVGD on MaF1.

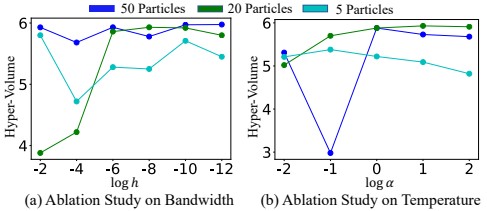

(a) Training  (b) Testing

Figure 4: (a) The Pareto front of classification loss and fairness penalty in the training set. MOO-SVGD obtains a uniform solution set that substantially dominates the baselines. (b) Trade-off between classification accuracy and CV score in the test set. PF-SMG ($k$ point) denotes PF-SMG with $k$ solution.

(a) Ablation Study on Bandwidth  (b) Ablation Study on Temperature

Figure 5: Ablation study of MOO-SVGD on ZDT1. (a) The final hypervolume w.r.t the bandwidth $h$ when $\alpha = 1$. The optimal $h$ is larger for smaller number of particles. (b) The final hypervolume w.r.t. the temperature $\alpha$ when $h = 1e - 5$. All the results are averaged over 3 runs.

**Trade-off Between Accuracy and Fairness** When a ML system is applied to a real-world scenario, it may be biased against people with sensitive attributes, e.g., their genders or races. This leads to unfairness in ML, which is a key issue in real applications of ML. A common practice to avoid unfairness is to add fairness regularization during training. Unfortunately, introducing fairness regularization may hurt the prediction accuracy of the model. The user may have various requirements for the trade-off between accuracy and fairness in different environments. This can be framed into a MOO problem, which simultaneously optimize the typical training loss and the fairness penalties.

For this experiment, please refer to the appendix to see how we construct our dataset. We choose *gender* to be the sensitive attribute, and we use *disparate impact* as the fairness criterion. Disparate impact can be measured by the CV score [2], $\text{CV}(f) = |p(\hat{y} = 1 | A = 0) - p(\hat{y} = 0 | A = 0)|$, where $f$ is our classifier, $\hat{y}$ is the prediction, and $A$ is the sensitive attribute. $\hat{y} = 1$ if $f(x) > 0$; otherwise, $\hat{y} = 0$. CV score implies the gap between the probabilities of getting positive outcomes in different sensitive groups. As in [23], we use a linear classifier with 87 parameters. The classification loss $\ell_1$ is the typical binary cross-entropy loss. For fairness penalty, instead of directly optimizing the CV score, we use a convex surrogate of CV score provided in [23], $\ell_2(f) = \left[\mathbb{E}_{x_i, a_i \in \mathcal{D}}(a_i - \bar{a})f(x_i)\right]^2$, where $x_i$ is the feature of instance $i$; $a_i$ is the sensitive attribute of instance $i$; $\mathcal{D}$ is the dataset, and $\bar{a}$ is the averaged value of all $a_i$ in the dataset. Basically, it is minimizing the empirical covariance between the the prediction and the sensitive attribute.

We compare with PF-SMG [23] and EPO [27]. The number of particles is 10 for MOO-SVGD and EPO. For PF-SMG, we use two different versions. The first version has a maximal particle number of 600; the second is 50. The two versions are denoted by PF-AMG-50 and PF-SMG-600. The results are shown in Fig. 4. Generally, MOO-SVGD yields the best solution set at convergence. Among all

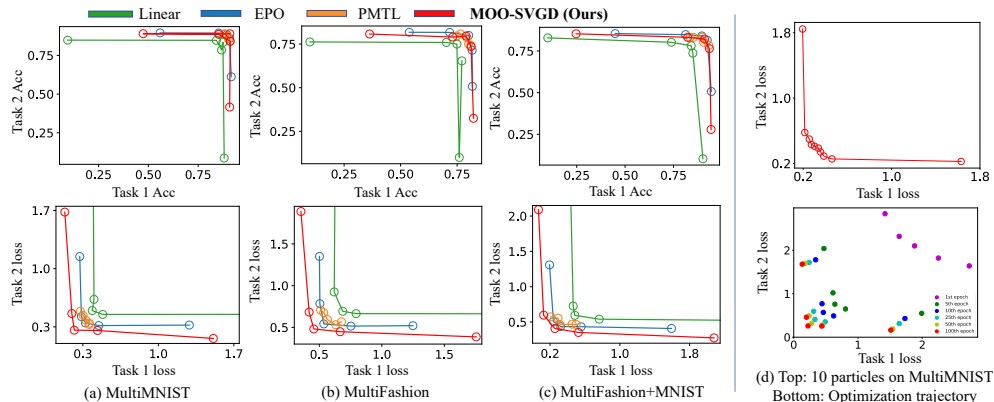

Figure 6: The solution set of the neural networks obtained by different methods on two tasks, w.r.t. the training loss (lower panel) and testing accuracy (upper panel). MOO-SVGD yields the best Pareto front on training loss and a widespread Pareto front on test accuracy.

the methods, only MOO-SVGD and PF-SMG-600 can get a solution set that is uniformly distributed on different preferences. PF-SMG-50, due to poor exploration ability of the random strategy, shrinks to the two end points of PF-SMG-600. EPO totally fails in this task. Uniformly sampling preference vector from the semi-circle is problematic, since the Pareto front lies in a very small region of the semi-circle, and most of the preference vectors do not ends up with an 'exact' solution at all.

**Multi-task Learning with Neural Networks** MultiMNIST, MultiFashion and MultiFashion+MNIST, introduced in [34], are three standard benchmarks for multi-task deep learning. The datasets are created in a similar way: each image in the datasets is a combination of two different images. One of them is on the top-left, and the other is on the bottom-right. In MultiMNIST, both images are from the MNIST dataset [16]. Similarly, MultiFashion uses two images from the FashionMNIST dataset [38]. For MultiFashion+MNIST, the top-left image is from MNIST, while the bottom-right one is from FashionMNIST. All three datasets have 120,000 samples in the training set, and 20,000 samples for test set. These datasets are used in [18, 27, 26].

The baseline methods are linear scalarization, EPO [27] and Pareto Multi Task Learning (PMTL) [18]. EPO and PMTL both require preference vectors to work. We follow the setting in [27], where the authors use LeNet [16] and 5 preference vectors uniformly sampled from a semi-circle. We also use 5 particles for MOO-SVGD. We find that training with MOO-SVGD can be problematic in the early stage, where networks with different initialization can be too close to each other and cause gradient explosion. Therefore, we initialize the networks with same parameters, then warm-up them by training with naive linear scalarization objective for one epoch, where the weights are sampled uniformly from $[0, 1]$. We switch back to MOO-SVGD in the remaining epochs. For all the experiments, all the methods are trained for 100 epochs with a batch size of 256. Moreover, they are optimized with SGD and the learning rate is $0.001$.

## 6 Conclusion, Limits, Impacts, and Open Questions

In this paper, we propose Multi-objective Stein Variational Gradient Descent (MOO-SVGD). Empirically, it can trace the Pareto front evenly without user preference. There are a number of limits that deserve future efforts. On the practical side, since our methods leverage SVGD and Langevin dynamics, they necessarily inherent their limits, such as the sensitivity on the choice of kernel and step sizes; studies on automatizing the selection of these hyper-parameters can improve the method significantly. Theoretically, analyzing our methods are challenging due to the lack of explicit form of the invariant distributions. For example, although we show that the limiting distributions concentrate on the Pareto front, it is an open question to theoretically characterize in what sense they are evenly distributed in the Pareto front. Moreover, it would be interesting to draw theoretical understandings on why the MOO-SVGD performs better than the MOO-Langevin dynamics.

In terms of social impact, as a new computational tool for MOO, it allows us to better trade-off different objectives in machine learning and hence improve ML systems in both accuracy and various trustworthy metrics such as fairness, safety, as we demonstrate with our experiments. We will release code online to promote the applications of our approach, although it does open the possibility of malicious use of our techniques for adversarial purposes.

## Acknowledgements

The work is conducted in the statistical learning and AI group in computer science at UT Austin, which is supported in part by CAREER-1846421, SenSE-2037267, EAGER-2041327, and Office of Navy Research, and NSF AI Institute for Foundations of Machine Learning (IFML). Xingchao Liu is supported in part by a funding from BP. X. T. Tong's research is supported by Singapore MOE Academic Research Funds R-146-000-292-114. The authors thank the reviewers for all the suggestions made in the reviewing process.

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
