# A  Proofs

We begin our discussion by the following result regarding the derivatives:

**Lemma A.1.** *The following holds under Assumption 1:*

1. *Given $\lambda \in \mathbb{R}^m$ so that $\sum_{i=1}^m \lambda_i = 1$ and $\lambda_i \geq 0$, we denote $F_\lambda = \sum_{i=1}^m f_i$ and $G_\lambda = \|\nabla F_\lambda\|^2$. Then the gradient satisfies*
$$g_\lambda^\top \nabla G_\lambda \geq c_1 G_\lambda, \quad \|\nabla G_\lambda\|^2 \leq c_2^2 G_\lambda.$$
   *And for any $\alpha > 0$, the Laplacian satisfies*
$$\Delta G_\lambda \leq \frac{c_1}{2\alpha} G_\lambda + 2Md$$
   *where $M = c_3^2 \alpha / c_1 + c_2^2$.*

2. *If $G^*(x) = \|g^*(x)\|^2$ is $C^1$, then*
$$\|\nabla G^*(x)\| \leq c_2 \|g^*(x)\|, \quad g^*(x)^\top \nabla G^*(x) \geq c_1 G^*(x).$$

*Proof.* **Claim 1.** Note $\nabla G_\lambda = 2\nabla^2 F_\lambda g_\lambda$ and that
$$\frac{1}{2} c_1 I \preceq \nabla^2 F_\lambda = \sum_{i=1}^m \lambda_i \nabla^2 f_i \preceq \frac{1}{2} c_2 I.$$
We immediately obtain the first result regarding $\nabla G_\lambda$. Next, we check the components of derivatives:
$$[\nabla G_\lambda]_i = 2\sum_j [\nabla^2 F_\lambda]_{i,j} [g_\lambda]_j,$$
so
$$
\begin{aligned}
\Delta G_\lambda &= \sum_i [\nabla^2 G_\lambda]_{i,i} \\
&= 2\sum_i \sum_j [\partial_{i,i,j}^3 F_\lambda][g_\lambda]_j + 2\sum_i \sum_j (\partial_{i,j} F_\lambda)^2 \\
&\leq 2c_3 \sqrt{d} \|g_\lambda\| + 2d\|\nabla^2 F_\lambda\|^2 \\
&\leq \frac{c_1}{2\alpha^2} \|g_\lambda\|^2 + 2c_3^2 d\alpha/c_1 + 2dc_2^2.
\end{aligned}
$$
**Claim 2.** Consider an arbitrary vector $v$, we need to show for small enough $\epsilon$,
$$-\epsilon c_2 \|g^*(x)\| \leq G^*(x + \epsilon v) - G^*(x) \leq \epsilon c_2 \|g^*(x)\|.$$
To do so, let $\beta = \lambda(x), \gamma = \lambda(x + \epsilon v)$. Then by Taylor expansion, there is a $w$ such that
$$G^*(x + \epsilon v) - G^*(x) \leq G_\beta(x + \epsilon v) - G_\beta(x) = \epsilon v^\top \nabla G_\beta(x) + \frac{1}{2}\epsilon^2 v^\top \nabla^2 G_\beta(w)v.$$
Note that $\nabla G_\beta(x) = 2\nabla^2 F_\beta(x) g_\beta(x)$ and $\nabla^2 F_\beta \preceq \frac{1}{2}c_2 I$. Letting $v = \nabla G_\beta/\|\nabla G_\beta\|, \epsilon \to 0$, we find that
$$\lim_{\epsilon \to 0} \frac{1}{\epsilon}(G^*(x + \epsilon v) - G^*(x)) \leq \|\nabla G_\beta(x)\| \leq c_2 \|g^*(x)\|.$$
Likewise, we have
$$G^*(x + \epsilon v) - G^*(x) \geq G_\gamma(x + \epsilon v) - G_\gamma(x) = \epsilon v^\top \nabla G_\gamma(x) + \frac{1}{2}\epsilon^2 v^\top \nabla G_\gamma(w)v.$$
From this we find that
$$\lim_{\epsilon \to 0} \frac{1}{\epsilon}(G^*(x + \epsilon v) - G^*(x)) \geq -\|\nabla G_\gamma(x)\|^2 \geq -c_2 \|g^*(x)\|.$$
The second part can be obtained if we let $v = g^*(x)$, and note that
$$G^*(x + \epsilon v) - G^*(x) \leq G_\beta(x + \epsilon v) - G_\beta(x) = -2\epsilon v^\top \nabla^2 F_\beta v + \frac{1}{2}\epsilon^2 v^\top \nabla^2 G_\beta(w)v.$$
Since $\nabla^2 F_\beta \succeq \frac{1}{2} c_1 I$, divide both sides with $\epsilon$ and let $\epsilon \to 0$, we find that
$$g^*(x)^\top \nabla G^*(x) \leq -c\|g^*(x)\|^2 = -cG^*(x).$$

$\square$

*Proof of Theorem 3.1.* Define $\jmath(x) = g^*(x) + \alpha\nabla\log\rho^*(x)$. Applying Stein's identity, we have

$$\phi_{\rho^*}^*(\cdot) = \mathbb{E}_{x\sim\rho^*}[g^*(x)k(x,\cdot) - \alpha\nabla_x k(x,\cdot)]$$
$$= \mathbb{E}_{x\sim\rho^*}[(g^*(x) + \alpha\nabla_x\log\rho^*(x))\log k(x,\cdot)]$$
$$= \mathbb{E}_{x\sim\rho^*}[\jmath(x)k(x,\cdot)]$$
$$= \jmath_{[\rho^*]}(\cdot).$$

Therefore, as $\rho^*$ is the fixed point of (4), we have

$$\nabla_x\cdot\left(\jmath_{[\rho^*]}(x)\rho^*(x)\right) = \nabla_x\cdot\left(\phi_{g^*}^*(x)\rho^*(x)\right) = 0.$$

Then we have

$$\int\nabla_x\cdot\left(\jmath_{[\rho^*]}(x)\rho^*(x)\right)h(x)dx = 0.$$

Using integration by parts, we have

$$\int(\jmath_{[\rho^*]}(x)\rho^*(x))^\top\nabla h(x)dx = \mathbb{E}_{\rho^*}[\nabla h(x)^\top k(x,y)\jmath(x)] = 0.$$

And hence

$$\langle\jmath_{[\rho^*]},\ \nabla h_{[\rho^*]}\rangle_{\mathcal{H}} = \mathbb{E}_{(x,y)\sim\rho^*\times\rho^*}[\nabla h(x)^\top k(x,y)\jmath(x)]$$
$$= \int(\jmath_{[\rho^*]}(x)\rho^*(x))^\top\nabla h(x)dx$$
$$= -\int\nabla_x\cdot\left(\jmath_{[\rho^*]}(x)\rho^*(x)\right)h(x)dx$$
$$= 0.$$

$\square$

As a remark, using the Krylov–Bogoliubov existence theorem (see Corollary 11.8 of [6]), fixed points to (4) exist as long as one can show $\{\rho_t, t \geq 0\}$ is tight. Tightness of $\{\rho_t, t \geq 0\}$ can often be established if we have a uniform upper bound for $\|g_{[\rho_t]}^*\|_{\mathcal{H}}^2$, assuming $g$ is coercive and smooth. Such an upper bound can be found in Theorem 3.2. On the other hand, there is no guarantee to the uniqueness of $\rho^*$. In fact, the non-gradient component of $g^*$ can set up a divergence-free rotation, along which $\rho_t$ will form a limit-cycle. This is different from the Langevin dynamics.

*Proof of Theorem 3.2.* Simply note that by the fact that $\rho^*$ is a fixed point of (4),

$$0 = -\int\nabla\cdot(\phi_{\rho^*}^*\rho^*)(x)G^*(x)dx$$
$$= \int\phi_{\rho^*}^*(x)^\top\nabla G^*(x)\rho^*(x)dx$$
$$= \int\int k(x,y)\nabla G^*(x)^\top(g^*(y) + \frac{\alpha}{\sigma^2}(y-x))\rho^*(x)\rho^*(y)dxdy. \tag{11}$$

Then note that $k(x,y) = 0$ if $\|y - x\| \geq m_d\sigma$. And if $\|y - x\| \leq m_d\sigma$, we apply Lemma A.1 claim 2,

$$\nabla G^*(x)^\top(g^*(y) + \frac{\alpha}{\sigma^2}(y-x))$$
$$= \nabla G^*(x)^{*}{}^{(x)+\nabla G(x)^\top(g^*(y)-g^*(x)+\frac{\alpha}{\sigma^2}(y-x))}$$
$$\geq c_1 G^*(x) - (L + \alpha/\sigma^2)\|\nabla G^*(x)\|\|y - x\|$$
$$\geq \frac{1}{2}c_1 G^*(x) - \frac{1}{2c_1}(L + \alpha/\sigma^2)^2 m_d^2\sigma^2. \tag{12}$$

Therefore, (11) leads to

$$\int G^*(x)\rho^*(x)k(x,y)\rho^*(y)dxdy \leq \frac{1}{2c_1^2}(L + \alpha/\sigma^2)^2 m_d^2\sigma^2\int\rho^*(x)k(x,y)\rho^*(y)dxdy.$$

To reach our first claim, simply note that $G^*(x) + G^*(y) \geq 2g^*(x)^\top g^*(y)$, then because $k(x,y) = k(y,x)$,

$$\int G^*(x)\rho^*(x)k(x,y)\rho^*(y)dxdy \geq \int g^*(x)^\top g^*(y)\rho^*(x)k(x,y)\rho^*(y)dxdy = \|g^*_{[\rho^*]}(x)\|^2_{\mathcal{H}}.$$

For the second claim, we use inequality (12) again,

$$\frac{d}{dt}\mathbb{E}^{\rho_t}G^*(X) = \int \nabla \cdot (\phi^*_{\rho_t}\rho_t)(x)G^*(x)dx$$

$$= \mathbb{E}_{x,y\sim\rho_t}[-g^*(y)k(x,y)\nabla G^*(x) + \nabla_y k(x,y)\nabla G^*(x)]$$

$$= \mathbb{E}_{x,y\sim\rho_t}[-g^*(y)k(x,y)\nabla G^*(x) + \alpha(\frac{x-y}{\sigma^2})^\top \nabla G^*(x)k(x,y)]$$

$$\leq \mathbb{E}_{x,y\sim\rho_t}[-\frac{1}{2}c_1 G^*(x)k(x,y) + \frac{1}{2c_1}(L+\alpha/\sigma^2)^2 m_d^2\sigma^2 k(x,y)]$$

$$\leq \mathbb{E}_{x,y\sim\rho_t}[-\frac{1}{2}c_1 g^*(x)(g^*(y))^{(}x,y) + \frac{1}{2c_1}(L+\alpha/\sigma^2)^2 m_d^2\sigma^2 k(x,y)]$$

$$= -\frac{1}{2}c_1\|g^*_{[\rho_t]}(x)\|^2_{\mathcal{H}} + \frac{1}{2c_1}(L+\alpha/\sigma^2)^2 m_d^2\sigma^2\|1_{[\rho_t]}\|^2_{\mathcal{H}}$$

Therefore

$$\min_{s\leq t} -\frac{1}{2}c_1\|g^*_{[\rho_s]}(x)\|^2_{\mathcal{H}} + \frac{1}{2c_1}(L+\alpha/\sigma^2)^2 m_d^2\sigma^2\|1_{[\rho_s]}\|^2_{\mathcal{H}}$$

$$\leq \frac{1}{t}\int_{s=0}^\top \left(-\frac{1}{2}c_1\|g^*_{[\rho_s]}(x)\|^2_{\mathcal{H}} + \frac{1}{2c_1}(L+\alpha/\sigma^2)^2 m_d^2\sigma^2\|1_{[\rho_s]}\|^2_{\mathcal{H}}\right)ds \leq \frac{1}{t}\mathbb{E}^{\rho_0}G^*(x).$$

$\square$

*Proof of Theorem 3.3.* Since $\rho^*$ is a fixed point of (8), we find that

$$0 = \nabla \cdot (g^*(x)\rho^* + \alpha\nabla\rho^*(x)) = \nabla \cdot (\jmath(x)\rho^*(x)).$$

Then for any $h(x)$, we find our first claim using integration by parts

$$0 = \int h(x)\nabla \cdot (\jmath(x)\rho^*(x))dx$$

$$= -\int \nabla h(x)^\top \jmath(x)\rho^*(x)dx.$$

Next we check the KL-divergence

$$\frac{d}{dt}\mathrm{KL}(\rho_t \| \rho^*) = \int (\nabla\log\rho^*(x) - \nabla\log\rho_t)^\top(g^*(x) + \alpha\nabla\log\rho_t(x))\rho_t(x)dx$$

$$= \int (\nabla\log\rho^*(x) - \nabla\log\rho_t)^\top(-\alpha\nabla\log\rho^*(x) - \alpha\jmath(x) + \alpha\nabla\log\rho_t(x))\rho_t(x)dx$$

$$= -\alpha F(\rho_t\|\rho^*) - \alpha\int (\nabla\log\rho^*(x) - \nabla\log\rho_t)^\top \jmath(x)\rho_t(x)dx.$$

Finally, we note that if we denote $h(x) = \rho_t(x)/\rho^*(x)$, then

$$\int (\nabla\log\rho^*(x) - \nabla\log\rho_t)^\top \jmath(x)\rho_t(x)dx = -\int (\nabla\log h(x))^\top \jmath(x)h(x)\rho^*(x)dx$$

$$= -\int \nabla h(x)^\top \jmath(x)\rho^*(x)dx = 0.$$

Therefore we have our second claim.

$\square$

*Proof of Theorem 3.4.* We will only prove Claim 2. Then Claim 1 comes as a result when $t \to \infty$. Note that $\rho_t$ is the density of the diffusion process

$$dx_t = -g^*(x_t)dt + \sqrt{2\alpha}dw_t$$

where $w_t$ is the Brownian motion. Next, we let $\lambda^*(x)$ be the solution to problem (2) and $\lambda_t = \lambda^*(x_t)$. Note that when applying the infinitesimal generator of $x_t$ to $G^*$

$$\begin{aligned}
\mathcal{L}G^*(x_t) :&= \lim_{\epsilon \to 0} \frac{1}{\epsilon} \left( \mathbb{E}(G^*(x_{t+\epsilon})|x_t) - (G^*(x_t)) \right) \\
&= \lim_{\epsilon \to 0} \frac{1}{\epsilon} \left( \mathbb{E}(G^*(x_{t+\epsilon})|x_t) - (G_{\lambda_t}(x_t)) \right) \\
&\leq \lim_{\epsilon \to 0} \frac{1}{\epsilon} \left( \mathbb{E}(G_{\lambda_t}(x_{t+\epsilon})|x_t) - (G_{\lambda_t}(x_t)) \right) \\
&= \mathcal{L}G_{\lambda_t}(x_t).
\end{aligned}$$

To continue, we use the Itô's formula and Lemma A.1 Claim 1,

$$\mathcal{L}G^*(x_t) \leq \mathcal{L}G_{\lambda_t}(x_t) = -\langle G_{\lambda_t}(x_t), g_{\lambda_t}(x_t) \rangle + \alpha \Delta G_{\lambda_t}(x_t)$$

$$\leq -\frac{1}{2}c_1 G^*(x_t) + 2\alpha Md.$$

So by Dynkin's formula formula

$$\mathbb{E}G^*(x_t) \leq \exp(-\frac{1}{2}c_1 t)\mathbb{E}G^*(x_0) + 4\alpha Md/c_1.$$

Next we investigate moment generating functions

$$V_\lambda(x) = \exp(bG_\lambda(x)), \quad V^*(x) = \exp(bG^*(x))$$

Then for $a \geq 0$, we have that

$$\begin{aligned}
\mathcal{L}V^*(x_t) &\leq \mathcal{L}V_{\lambda_t}(x_t) \\
&= b\langle \nabla G_{\lambda_t}, g_{\lambda_t} \rangle V_{\lambda_t}(x_t) + \alpha b(b\|\nabla G_{\lambda_t}\|^2 + \Delta G_{\lambda_t})V_{\lambda_t}(x_t) \\
&\leq (-c_1 bG_{\lambda_t} + \alpha b^2 c_2^2 G_{\lambda_t} + \frac{c_1}{2}bG_{\lambda_t} + 2\alpha bMd)V_{\lambda_t}(x_t)
\end{aligned}$$

When

$$b \leq \frac{c_1}{4\alpha c_2^2},$$

we have a further upper bound

$$\begin{aligned}
\mathcal{L}V^*(x_t) &\leq (-\frac{c_1 b}{4}G_{\lambda_t}(x_t) + 2\alpha bMd)V_{\lambda_t}(x_t) \\
&= (-\frac{c_1 b}{4}G^*(x_t) + 2\alpha bMd)V^*(x_t).
\end{aligned}$$

To continue, we note that if $G^*(x_t) \geq 16\alpha Md/c_1$, then

$$\mathcal{L}V^*(x_t) \leq -\alpha bMdV^*(x_t).$$

If $G^*(x_t) \leq 16\alpha Md/c_1$, then

$$\mathcal{L}V^*(x_t) \leq -\alpha bMdV^*(x_t) + 3\alpha bMdV^*(x_t) \leq -\alpha bMdV^*(x_t) + 3\alpha^2 bMd\exp(16\alpha bMd/c_1).$$

Therefore, in both cases, we have that

$$\mathcal{L}V^*(x_t) \leq -\alpha bMdV^*(x_t) + 3\alpha bMd\exp(16\alpha bMd/c_1).$$

So by Dynkin's formula, we find that

$$\mathbb{E}V^*(x_t) \leq \exp(-\alpha bMdt)\mathbb{E}V^*(x_0) + 3\exp(16\alpha bMd/c_1).$$

$\square$

# B   Additional Experiment Details

We provide additional experiment settings and details in this section.

## B.1   Algorithm Configuration

We use Adam [14] optimizer for MOO-SVGD in all the experiments. The learning rate is set differently for each task. An important aspect of MOO-SVGD is the choice of kernel. In standard SVGD, the authors use the RBF kernel, which measures the similarity between the particle $x_i$ and $x_j$ and penalize the pairs with small distances. However, in MOO, we want the objective vectors of the particles, $F(x_i)$, to be diverse. A large distance between $x_i$ and $x_j$ does not necessarily imply a large distance between $F(x_i)$ and $F(x_j)$. Therefore, we apply the RBF kernel on the function values, yielding $k(x_i, x_j) = \exp\left(-\frac{||F(x_i) - F(x_j)||^2}{h^2}\right)$, where $h$ is a hyperparamter called bandwidth. We use the median trick as in [20] and set $h = c \times \mathrm{med}$, where $\mathrm{med}$ is the median of the pairwise distances of $\{F(x_i)\}$ and $c$ is a small constant chosen from $\{10^{-2}, 10^{-3}, \ldots, 10^{-8}\}$. For each task, $c$ is chosen to maximize the resulting HV indicator.

For our experiment, we adopt the setting in [23]. The experiment is performed on the *Adult Income* dataset, which contains 30,162 training samples and 15,060 test samples. It is a binary classification problem, whose prediction target is whether the income of a person is higher than 50,000 dollars per year. Following [23], we randomly sample a subset of 5,000 data points from the training set as our training set. Each data point has a 86-dimensional feature.

## B.2   ZDT Problems

ZDT problems are 30-dimensional optimization problems. Assume the variable is $x = (x_1, x_2, \ldots, x_{30})$. All of problems are constructed in the following way,

$$
\begin{aligned}
\min \quad & f_1(x), \\
\min \quad & f_2(x) = g(x) \cdot h(f_1(x), g(x)), \\
s.t. \quad & 0 \leq x_i \leq 1, i \in \{1, 2, \ldots, 30\}.
\end{aligned}
\tag{13}
$$

$f_1(x), g(x)$ and $h(f_1(x), g(x))$ are different for different problems. The Pareto front of ZDT problems can be analytically solved. To deal with the constraint, we perform projected gradient descent, which means that we clip value of the variable back to $[0, 1]^{30}$ after each gradient descent step.

**ZDT1**   ZDT1 problem is defined by the following functions,

$$
f_1(x) = x_1, \quad g(x) = 1 + \frac{9}{29}\sum_{i=2}^{30} x_i, \quad h(f_1, g) = 1 - \sqrt{f_1/g}.
\tag{14}
$$

The Pareto optimal solutions can be characterised by,

$$
0 \leq x_1^* \leq 1 \quad \text{and} \quad x_i^* = 0 \quad \text{for} \quad i \in \{2, \ldots, 30\}.
\tag{15}
$$

**ZDT2**   ZDT2 problem is defined by the following functions,

$$
f_1(x) = x_1, \quad g(x) = 1 + \frac{9}{29}\sum_{i=2}^{30} x_i, \quad h(f_1, g) = 1 - (f_1/g)^2.
\tag{16}
$$

The Pareto optimal solutions can be characterised by,

$$
0 \leq x_1^* \leq 1 \quad \text{and} \quad x_i^* = 0 \quad \text{for} \quad i \in \{2, \ldots, 30\}.
\tag{17}
$$

**ZDT3**   ZDT3 problem is defined by the following functions,

$$
f_1(x) = x_1, \quad g(x) = 1 + \frac{9}{29}\sum_{i=2}^{30} x_i, \quad h(f_1, g) = 1 - \sqrt{f_1/g} - (f_1/g)sin(10\pi f_1).
\tag{18}
$$

The Pareto optimal solutions can be characterised by,

$$
S = [0, 0.0830] \cup [0.1822, 0.2577] \cup [0.4093, 0.4538] \cup [0.6183, 0.6525] \cup [0.8233, 0.8518]
$$
$$
\text{and} \quad x_i^* = 0 \quad \text{for} \quad i \in \{2, \ldots, 30\}.
\tag{19}
$$

**Hyper-parameter Configuration for MOO-SVGD** For ZDT1 and ZDT2, we use a learning rate of $5e-4$, and optimize the particles for 10,000 steps. The bandwidth constant $c = 1e-6$ and $\alpha = 1$. For ZDT3, we use a learning rate of $5e-5$, and optimize the particles for 10,000 steps.

**Hyper-parameter Configuration for MOO-LD** For all three ZDT problems, we use a step size of 0.1, and a temperature $\alpha = 0.01$. We sample 200,000 solutions and remove the dominated solutions from the solution set. The noise is zero in the first 20,000 steps to accelerate the process of reaching the Pareto front.

### B.3 MaF1 Problem

Assume $x = (x_1, x_2, \ldots, x_d)$ is a $d$-dimensional vector. The objectives for MaF1 are as follows,

$$f_1(x) = (1 - x_1 x_2)(1 + g(x)), f_2(x) = (1 - x_1(1 - x_2))(1 + g(x)), f_3(x) = x_1(1 + g(x)),$$

$$\text{where } g(x) = \sum_3^d (x_i - 0.5)^2, \text{ and } 0 \leq x_i \leq 1, i \in \{1, 2, \ldots, d\}.$$

(20)

In our experiments, we set $d = 10$. We perform projected gradient descent during optimization.

**Hyper-parameter Configuration for MOO-SVGD** We use a learning rate of $5e-3$, and optimize the particles for 10,000 steps. The bandwidth constant $c = 1e-2$ and $\alpha = 0.5$. We use 100 particles. We remove the dominated particles from the obtained solution set after convergence.

**Hyper-parameter Configuration for MOO-LD** We use a step size of $1e-3$, and a temperature $\alpha = 0.05$. We sample 200,000 solutions and remove the dominated solutions from the solution set. The noise is zero in the first 400 steps to accelerate the process of reaching the Pareto front.

The reference point for computing hypervolume indicator is $[1.0, 1.0, 2.0]$.

### B.4 Trade-off Between Accuracy and Fairness

In *Adult Income* dataset [2], each data point in the dataset has 14 features, including categorical features and numerical features. We expand all the categorical features into one-hot vectors, and normalize the numerical features to $[-1, 1]$, resulting in 86-dimensional features.

**Hyper-parameter Configuration for MOO-SVGD** We initialize our optimizer with a learning rate of $0.1$, and optimize the particles for 5,000 steps. We decrease the learning rate to $5e-2, 1e-2, 5e-3$ at the 500-th, 2000-th, and 3000-th step, respectively. The bandwidth constant $c = 1e-11$ and $\alpha = 1$. We use 10 particles.

## C Additional Experiment Results

We provide additional experiment results and visualization in this section.

### C.1 ZDT Problems in 300 seconds

We provide the results of ZDT problems in 300 seconds, an extended time limitation. Moreover, we add Bayesian-optimization algorithms for comparison, including DGEMO [25], USEMO-EI [1] and MOEA/D-EGO [39]. See Fig. 7. Note that direct comparison between white-box and bloack-box methods is unfair. The comparison here can just show gradient-based methods are more efficient when the functions are known, but generally white-box and black-box algorithms have different application scenarios.

### C.2 Training Dynamics

We provide the training dynamics of ZDT problems of PF-SMG, MOO-LD and MOO-SVGD for better understanding of the algorithms. See Fig. 12, Fig. 13 and Fig. 14.

---

[2]https://archive.ics.uci.edu/ml/datasets/adult

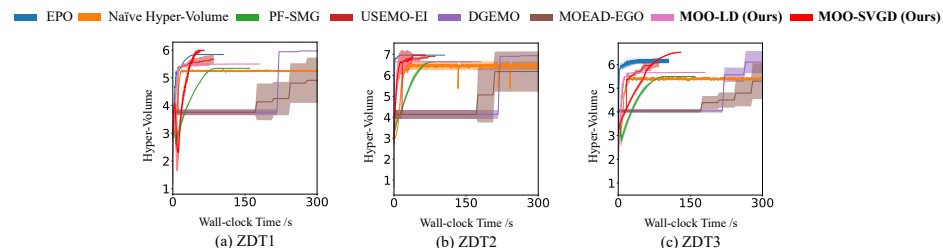

Figure 7: Runtime Comparison on ZDT Problems in 300 seconds

## C.3 Final results with different hyperparameters

We showcase the influence of different bandwidth and temperature by visualizing the final solution set of MOO-SVGD. See Fig. 8 and Fig. 9.

## D Discussion on Hypervolume Indicator & Its Optimization

Obviously, the HV indicator (Eq. (10)) can also be used as an objective function for optimizing solution sets. For example, [25, 7] greedily add new points to obtain the highest expected HV improvement. However, the landscape of the HV indicator is piece-wise constant (similar to the 0-1 loss in classification) and is difficult to optimize with gradient descent. Particularly, for all the dominated points in the solution set, their gradient is zero. Only the non-dominated points get non-zero gradients. Consequently, most of the particles will not be updated, and the solution set cannot cover the Pareto front. See Fig. 10.

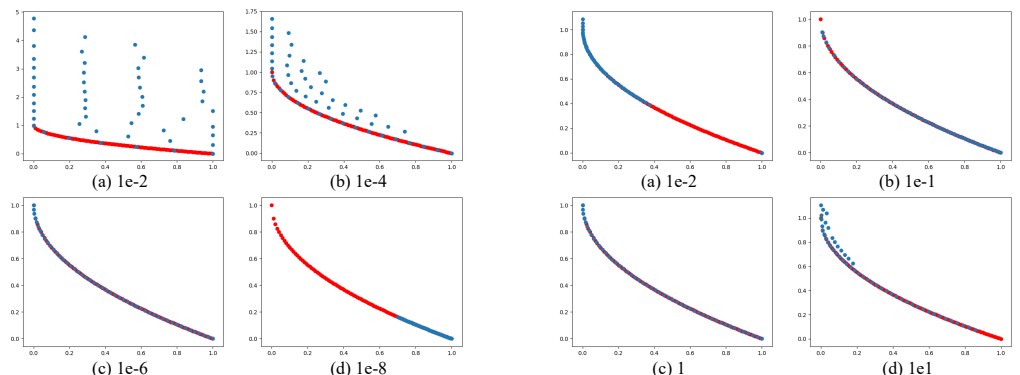

Figure 8: Solution Set of MOO-SVGD with Different Bandwidth

Figure 9: Solution Set of MOO-SVGD with Different temperature

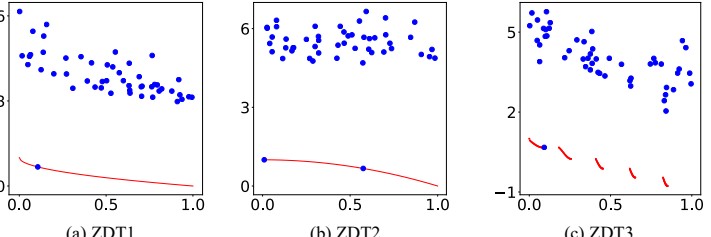

Figure 10: Direct Optimization with Hypervolume

## E Discussion on Hypervolume Indicator Gradient Ascent (HIGA)

HIGA [36] is also a gradient-based algorithm for find the whole Pareto front. To assign gradients to the dominated points (and hence enables gradient-based optimization), each particle in their algorithm simply ignores the other particles that dominate it. We implement HIGA by ourselves, and find that it

has a number of drawbacks compared with our method empirically: (1) the result of HIGA depends on the choice of the reference point (See Fig. 11(b) ), which need to be specified by the user based on an estimation of function range beforehand; in particular, if a particle excesses the range of the reference point, it would receive no gradient (See the two points at the top of Fig. 11(a) Iteration 1000). (2) We found that in HIGA, the particles can overlap on the Pareto front (See Fig. 11(a)). In comparison, in this case MOO-SVGD will move the particles apart with the repulsive force. HIGA is converged in all the experiments.

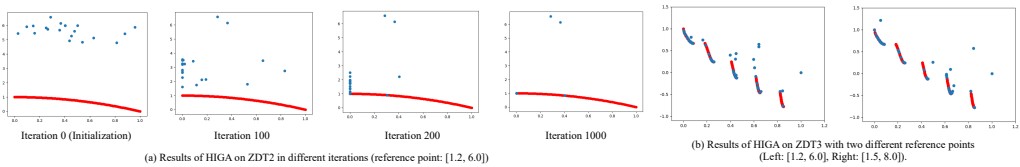

Figure 11: Experiment Results of HIGA on ZDT. Please zoom in to see the details.

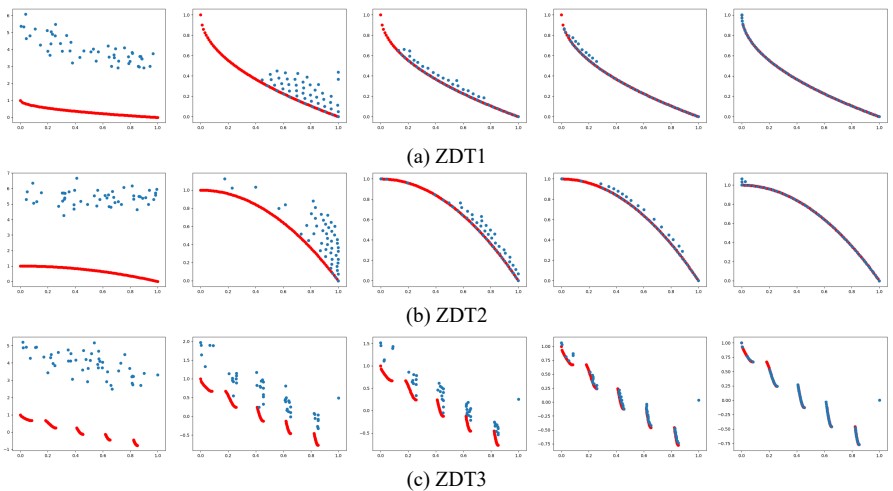

Figure 12: Optimization dynamics of MOO-SVGD

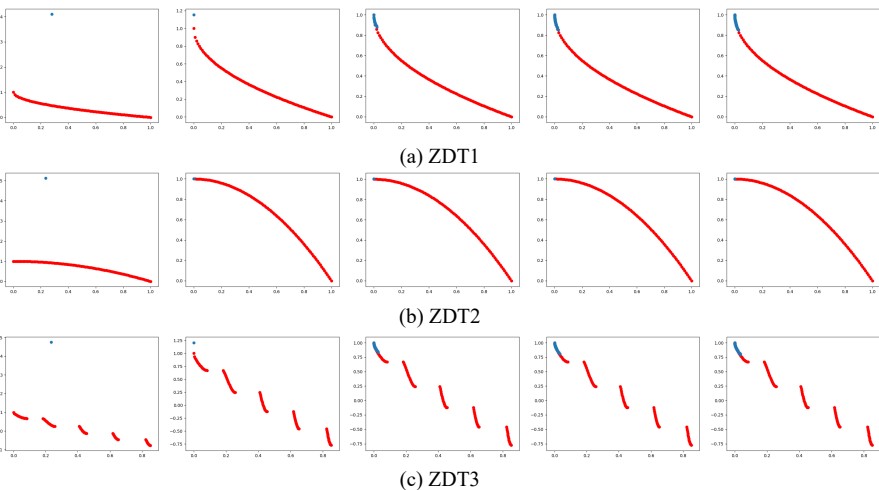

Figure 13: Optimization dynamics of MOO-LD

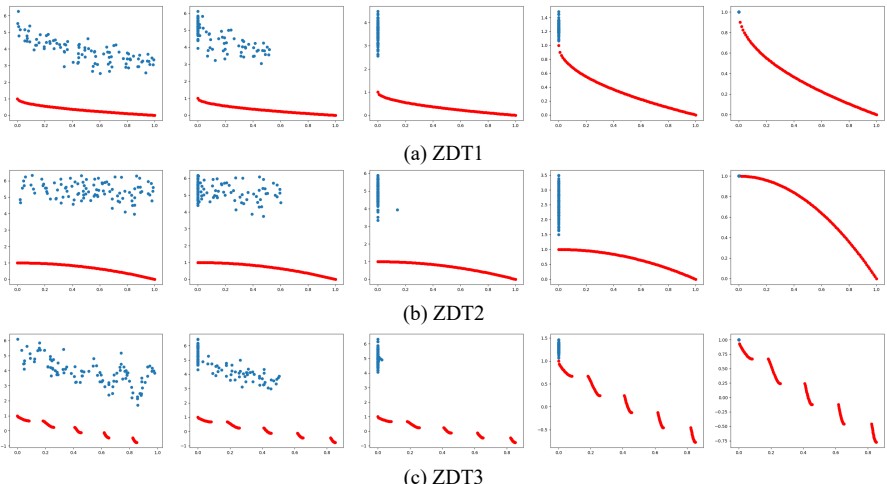

(a) ZDT1

(b) ZDT2

(c) ZDT3

Figure 14: Optimization dynamics of PF-SMG

# F  Discussion on Pareto hypernet methods

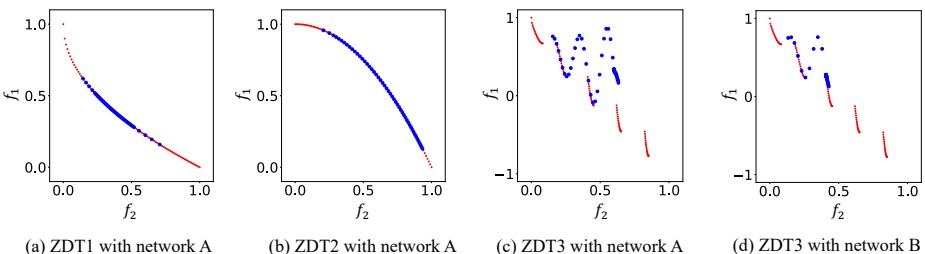

(a) ZDT1 with network A    (b) ZDT2 with network A    (c) ZDT3 with network A    (d) ZDT3 with network B

Figure 15: COSMOS on ZDT Problems. Network A is a two-layer ReLU network with 150 hddien neurons. Network B is a three-layer ReLU network with 60 neurons in each hidden layer.

We provide more discussion on Pareto hypernet methods and provide additional results of COSMOS with different network structures (including the original results on ZDT problems) in Fig. 15. From the ZDT experiments in the main text we find that:

(1) These methods still need to assign preference vectors manually (even though it is not needed theoretically if the network could be trained to be perfect), and a uniform assignment may not cause a uniform Pareto front, as we show in ZDT1 (Fig. 15 (a)) and ZDT2 (Fig. 15 (b)). In comparison, MOO-SVGD can generate an uniform Pareto front as we show in the paper.

(2) The methods based on hypernets may be limited by the optimization of the function approximator. In ZDT3 (Fig. 15 (c)), several points are not on the real Pareto front. In contrast, we show both theoretically and empirically that MOO-SVGD converges to the Pareto front.

We also find that: (3) Choosing appropriate hypernet structure can be tricky. In Fig. 15 (c) and Fig. 15 (d), we use a two-layer neural network and a three-layer neural network with similar number of parameters (4,800 vs. 5,520). We train both networks to convergence and find that they yield very different results. Note that MOO-SVGD only maintains 1,500 parameters to store the solutions in this setting.

# G  What causes the sub-optimal empirical performance of MOO-LD?

We assume there are two reasons for its empirically worse performance: (1) it requires longer burn-in time to effectively draw sample, and (2) the objectives in the experiments are non-convex. We run additional experiments to validate our conjectures. Results are shown in Fig. 16.

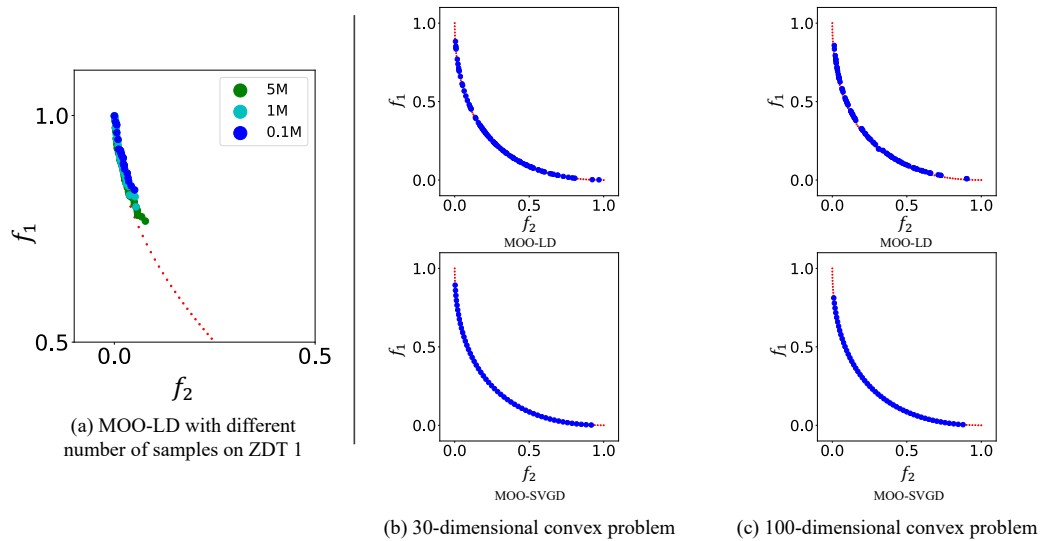

(a) MOO-LD with different number of samples on ZDT 1

(b) 30-dimensional convex problem

(c) 100-dimensional convex problem

Figure 16: Additional Experiments with MOO-LD

For (1), we run MOO-LD for 5 million iterations on ZDT1. We observe that the algorithm is slowly making progress with more iterations. See Fig. 16(a).

For (2), in the main text, we show that MOO-LD works on par with MOO-SVGD on a 1d toy example with convex objectives $f_1(x) = x^2$ and $f_2(x) = (1-x)^2$. We generalize the problem to dimensions, with $f_1(x) = \frac{1}{d}\sum_{i=1}^d x_i^2$ and $f_2(x) = \frac{1}{d}\sum_{i=1}^d (1-x_i)^2$. The results are shown in Fig. 16(b) and Fig. 16(c). We observe that MOO-LD still provide comparable Pareto front as MOO-SVGD even on 100-dimensional MOO problem. This verifies our theory which claims that MOO-LD converges to the true Pareto front for convex objectives, and also suggests the potential of MOO-LD. We leave the improvement of MOO-LD to high-dimensional non-convex problems to future works.