# OpenReview forum: "Profiling Pareto Front With Multi-Objective Stein Variational  Gradient Descent"
_NeurIPS.cc/2021/Conference — NeurIPS 2021 Spotlight_

### Official Review · Reviewer_1ANJ · 2021-07-01

**Rating:** 6
**Confidence:** 4

**Summary:**

The paper proposes novel gradient-based algorithms for finding solutions across the Pareto front by using Stein variational gradient descent (SVGD) (and Langevin dynamics), in multi-objective optimization (MOO) problems.


**Limitations And Societal Impact:**

Yes.

**Main Review:**

Strengths:
- The paper is well structured and well written.
- The paper focuses on the main challenge of MOO problems: find representative solutions on the Pareto front, and propose a novel and scalable algorithm for this problem.
- The general idea is well explained, and the motivation is clear and straightforward.
- The authors provide theoretical guarantees and analysis for the proposed method.

Questions/suggestions/Weaknesses:

- Overall it is a good paper with clear motivation and good theoretical analysis. I have two main concerns:
   - Previous literature in the same direction as discussed in the paper is omitted from discussion and empirical evaluation. Specifically, recent methods propose profiling the entire Pareto front in a scalable manner by conditioning the model on the preference direction (e.g. [1, 2]).
   - Second, scalability of the approach: the method essentially trains multiple models. The number of models that are needed for achieving good cover of the Pareto front grows exponentially with the number of objectives. This may be a major drawback with large-scale problems and models.

- Experiments:
   - Missing comparisons to recent state-of-the-art MOO approaches such as [1, 2].
   - The Linear scalarization baseline is missing from most comparisons.
   - The authors only experiment with small-scale and/or synthetic examples. I encourage the authors to provide an empirical evaluation on large-scale problems, like common MTL baselines (e.g., NUYv2/CityScapes dataset).
   - In addition, experiments with a larger number of objectives ($>3$) would be beneficial.
   - How did you select the hyper-parameters (HPs) for your method and baseline methods?
      - For example, in Appendix C.1, you state that $c$ (or the bandwidth) parameter is selected to optimize the hyper-volume. However, you did not use a validation split, so does this mean $c$ is optimized according to the test set? How did you select other HPs like $\alpha$?
      - All methods use the same learning rate (lr) in a given experimental setup if I understand correctly. How did you select this lr? Is it possible baseline methods can be improved by selecting an appropriate learning rate for each method?


[1] Learning the Pareto Front with Hypernetworks, Navon et al., ICLR 2021.

[2] Efficient Multi-Objective Optimization for Deep Learning, Ruchte et al., 2021.






**Time Spent Reviewing:**

4-5 hours

---

> ### Author Response · Authors · 2021-08-10
> **Authors' Response to Reviewer 1ANJ**
>
> **1. The Linear scalarization baseline is missing from most comparisons.**
>
> We have compared with EPO, which is expected to be better than linear scalaraization since it enforces uniform distribution on the ratios of the  different objectives, rather than the linear weights (see [1]). However, we agree that it is a good idea to include linear scalarization baseline for completeness. We will add them in the revision.
>
> [1] Debabrata Mahapatra and Vaibhav Rajan. Multi-task learning with user preferences: Gradient descent with controlled ascent in pareto optimization. In International Conference on Machine Learning, pages 6597–6607. PMLR, 2020.
>
> **2. More experiments with large-scale model and more objectives**
>
> Thanks for your suggestion. Our current work focuses on developing the novel algorithm and demonstrating its basic theoretical and empirical properties.  We are interested in doing more experiments on larger scale and state-of-the-art tasks, but this would require much more empirical efforts that may not be directly related to the introduction of the basic method as our current work focuses on. But we certainly think that this is a necessary next step to pursuit in future works.
>
> **3. Number of particles grows exponentially with objectives**
>
> You are right that  the number of particles needed to form cover the Pareto front (say in the sense of $\epsilon$-cover) grows exponentially as the number of objectives increases. However, covering a high-dimensional set is the hardness of the problem itself and does not depend on the algorithms.  For EPO and PF-SMG, they also need an exponential number of solutions to uniformly cover the Pareto set.
> Pareto hypernets may feel favorably, but there is no magic with using neural networks in terms changing the hardness of the problem. In fact, as the authors witnessed, the  Pareto Fronts obtained by  [2,3] in practice tend to be  narrow due to optimization difficulties, and even if we obtain a perfect hypernet, we will still need to draw exponential  number of particles out of it if we want to cover the set mathematically.
>
> On the other hand, we do not have really cover the whole Pareto set in a rigorous mathematical sense since as we see in practice, a small number of particles can already improve significantly over what we can do with a single solution from traditional single objective optimization.
>
> [2] Navon, A., Shamsian, A., Chechik, G., & Fetaya, E. (2020). Learning the Pareto front with hypernetworks. ICLR 2021.
>
> [3] Ruchte, M., & Grabocka, J. (2021). Efficient Multi-Objective Optimization for Deep Learning. arXiv preprint arXiv:2103.13392.
>
> **4. Hyper-parameter selection and learning rate selection**
>
> **Criterion of hyper-parameter selection:** For ZDT problems and MaF1, we simply choose the hyper-parameter by grid search that achieves the largest HV since they are optimization problems. For 'trade-off between accuracy and fairness' and MultiMNIST experiments, we select the hyper-parameters that maximize the HV of the training objectives. We do so because we do not have a separate validation split. We will clarify that in revision.
>
> **Learning rate selection:** Thank you for mentioning this issue. Here, we elaborate how we choose learning rate for different experiments. We will revise our manuscript to include the information.
>
> (1) For ZDT, MaF1 and the fairness problem, we select the learning rate of each algorithm to maximize the obtained hyper-volume on the training loss.
>
> (2) For the MultiMNIST experiments, we follow the optimization strategy in EPO for fair comparison. In this setting, higher learning rate does lead to better training losses in 100 epochs. Therefore, we stick to the setting in EPO, where they use SGD optimizer with a learning rate of $0.001$ and $0$ momentum. See L342-343.
>
> We want to point out that the sub-optimal performance of the baselines are caused by their design more than the choice of the learning rate. For example, PF-SMG does not have a mechanism to ensure that the final solutions will distribute evenly on the Pareto set (see discussion in Line 234-241). As we show in Appendix D, the result of PF-SMG may even collapse to a single solution.
> In comparison, MOO-SVGD explores the solutions with explicit repulsive force, avoiding the particles from collapse while keeping high efficiency. For EPO, its uniform assignment of the preference vector can be problematic, especially in ZDT3 and the fairness experiment. We  discussed this issue in L323-325 and the caption of Figure 1 in the main text.
> For the real-world experiments (fairness and MultiMNISTs), we attribute the better Pareto front obtained by MOO-SVGD to the repulsive force between the particles, which pushes the particles away from each other and accelerates training.

---

> > ### Comment · Reviewer_1ANJ · 2021-08-17
> > **After rebuttal**
> >
> > I would like to thank the authors for the detailed response. I have read the reviews and the responses provided by the authors, and I still have two concerns: (1) First, regarding the scalability of the approach to many objectives: I suggest the authors include some experiments with more than two objectives, and to discuss the limitations of the approach in that sense. (2) Second, regarding the optimization of HPs over the train set which could lead to suboptimal results and damage the comparison.
> >
> > Nevertheless, I would like to increase my score to 6. I encourage the authors to revise the submission with the additional results and take into account the comments from the reviewers.

---

> > > ### Author Response · Authors · 2021-08-18
> > > **Thank You & Authors' Response**
> > >
> > > Thank you for reading our response and increasing the rating!
> > >
> > > Regarding your concerns:
> > > (1) Currently we are working on large-scale experiments, and we will add additional results in the later revised version.
> > > (2) We agree that selecting HPs with a validation set is more reasonable, but we select the hyper-parameters with training set because we do not have a validation set in the existing experiments. However, it is still fair comparison since the HPs are selected using the training set for all the methods.
> > > (3) We will revise the manuscript following other comments.
> > >
> > > Thanks again for your valuable feedback!

---

### Official Review · Reviewer_WBcz · 2021-07-17

**Rating:** 7
**Confidence:** 4

**Summary:**

This paper proposes a method to learn multiple Pareto solutions for multi-objective optimization (MOO).
Specifically, the method iteratively updates a set of points by Stein variational gradient descent (SVGD) to encourage the diversity between the solutions. Another counterpart method based on Langevin dynamics is also provided.
The methods are theoretically guaranteed to converge to the Pareto front.
The experimental results demonstrate the superiority of the proposed SVGD type of method compared with EPO. The superiority is shown visually and by the hypervolume of the solutions.

**Limitations And Societal Impact:**

Yes.

**Main Review:**

Strengths:
1. The idea of encouraging the diversity between the solutions by SVGD is interesting.
2. The proposed methods are theoretically guaranteed to converge to the Pareto front.
3. The experimental results demonstrate the superiority of the proposed SVGD type of method compared with EPO. The superiority is shown visually and by the hypervolume of the solutions.

Weaknesses:
1. As a method to learn multiple Pareto solutions for MOO, the proposed method discussed no related works that learn multiple Pareto solutions, e.g., [1,2,3,4,5].
2. The proposed method cannot control the specific preference vector for each solution, while [1,2] can, suggesting a type of weakness of the proposed method.
3. The proposed method requires maintaining multiple solutions, which may have computation and storage problems, whereas [5] maintains one solution only, suggesting another type of weakness of the proposed method.
4. Since MOO-Langevin Dynamics seems inferior for every aspect, the necessity to present this method is not clear.

[1] Navon, A., Shamsian, A., Chechik, G., & Fetaya, E. (2020). Learning the Pareto front with hypernetworks. ICLR 2021.

[2] Lin, X., Yang, Z., Zhang, Q., & Kwong, S. (2020). Controllable Pareto multi-task learning. arXiv preprint arXiv:2010.06313.

[3] Shah, A., & Ghahramani, Z. (2016, June). Pareto frontier learning with expensive correlated objectives. In International Conference on Machine Learning (pp. 1919-1927). PMLR.

[4] Deist, T. M., Grewal, M., Dankers, F. J., Alderliesten, T., & Bosman, P. A. (2021). Multi-Objective Learning to Predict Pareto Fronts Using Hypervolume Maximization. arXiv preprint arXiv:2102.04523.

[5] Ruchte, M., & Grabocka, J. (2021). Efficient Multi-Objective Optimization for Deep Learning. arXiv preprint arXiv:2103.13392.

Originality: Are the tasks or methods new? Is the work a novel combination of well-known techniques? (This can be valuable!) Is it clear how this work differs from previous contributions? Is related work adequately cited?

1. The idea of encouraging the diversity between the solutions by SVGD is interesting.
2. No related work that learn multiple Pareto solutions is discussed.

Quality: Is the submission technically sound? Are claims well supported (e.g., by theoretical analysis or experimental results)? Are the methods used appropriate? Is this a complete piece of work or work in progress? Are the authors careful and honest about evaluating both the strengths and weaknesses of their work?

-Yes.

Clarity: Is the submission clearly written? Is it well organized? (If not, please make constructive suggestions for improving its clarity.) Does it adequately inform the reader? (Note that a superbly written paper provides enough information for an expert reader to reproduce its results.)

-Fair.

Significance: Are the results important? Are others (researchers or practitioners) likely to use the ideas or build on them? Does the submission address a difficult task in a better way than previous work? Does it advance the state of the art in a demonstrable way? Does it provide unique data, unique conclusions about existing data, or a unique theoretical or experimental approach?

-The experimental results demonstrate the superiority of the proposed SVGD type of method compared with EPO. The superiority is shown visually and by the hypervolume of the solutions.

**Time Spent Reviewing:**

5

---

> ### Author Response · Authors · 2021-08-10
> **Authors' Response to Reviewer WBcz**
>
> **1. Comparison with baselines & Handling preference vectors**
>
> Please refer to General Response #1.
>
> **2. Regarding MOO-LD**
>
> Please refer to General Response #2.
>
> **3. Additional computation and storage**
>
> Thank you for pointing that out. Like we said in General Response #1, each method has its pros and cons.
> One advantage of MOO-LD over MOO-SVGD is that it only needs to maintain one solution during training, but MOO-SVGD converges faster and get better results because it is a deterministic update and do not require a burn-in phase like MOO-LD.
> An interesting direction is to further speed up MOO-SVGD and reduce its overhead by combining MOO-LD, which we leave for future works.

---

> > ### Comment · Reviewer_WBcz · 2021-08-17
> > **Upgraded my score to 7**
> >
> > I appreciate the detailed response provided by the authors. I suggest the authors adding those things to the main paper or to the supplementary materials. I upgraded my score to 7.

---

> > > ### Author Response · Authors · 2021-08-18
> > > **Thank You & Authors' Response**
> > >
> > > Thank you for reading our response and increasing the score!
> > >
> > > We will complete the comparison with the baselines and refine the discussion on MOO-LD. We will add these contents to the later revised manuscript. Moreover, we will update the manuscript according to the other comments from the reviewers.
> > >
> > > Thanks again for your valuable comments and suggestions!

---

### Official Review · Reviewer_ag1C · 2021-07-17

**Rating:** 6
**Confidence:** 4

**Summary:**

This paper studies how to acquire diverse and representative Pareto solutions by gradient-based methods. Two algorithms -- MOO-SVGD and MOO-LD are proposed by using stein variational gradient descent and Langevin dynamics respectively, with convergence (to local Pareto optimal) guarantees. The experimental results demonstrate that MOO-SVGD can obtain visually uniform solution sets of ZDT and MaF1 problems, and get better Pareto fronts on fair ML and multi-task learning.


**Main Review:**

Strength:

The motivation is clear to the readers that the diversity of the Pareto solution set is crucial to MOO.
The intuitive review of the algorithm design is clear for both two algorithms.
Theoretical convergence of strongly convex functions is provided.
The experimental results are convincing and attractive, especially the ZDT and MaF1 problems.

Weakness:

Although the intuition behind using SVGD is clear, it is unclear that why MOO-SVGD is superior to find diverse Pareto solutions from a theoretical review (the authors show this limitation in Section 6).
Assumption 1 is too strong for the deep learning model, where the ReLu layer is not strongly convex, which is not sufficient to claim “we mainly focus on the finding locally optimal solutions” (line 68).
I think MOO-LD is not important to the paper logic, and the experiments show its impractical performance. What’s the purpose that the authors care about this? In the experiments, the authors also do not give an intuitive interpretation on why MOO-LD fails.
In the fair ML and multi-task learning experiments, the solutions are not that uniform as the previous 2 experiments. Could the authors provide more explanations?

Typos:
Line 168, “j,j,k” under the partial differential symbol.
Line 291, Fig 5 → Fig 2


**Time Spent Reviewing:**

48

---

> ### Author Response · Authors · 2021-08-10
> **Authors' Response to Reviewer ag1C**
>
> **1. Regarding Assumption 1**
>
> You are right that Neural Networks are non-convex. But this is also a standing issue even for optimization theory with only one objective function.  Analysis for MOO convergence is much harder than single objective. We agree that our phrase in the text may lead to  confusion, so we have removed that statement with our discussion of locally optimal Pareto solution to avoid confusion.
>
> **2. Regarding MOO-LD**
>
> Please refer to General Response #2.
>
> **3. Why not that uniform in the last two experiments?**
>
> We attribute this phenomenon to early-stopping. Referring to Appendix D.2, MOO-SVGD tends to first achieve the true Pareto front, then start to spread out. However, in our experiments, the models are not fully optimized to avoid overfitting. Our experiments demonstrates that MOO-SVGD can also give diverse solutions with early-stopping (though not so uniform), which is usual in modern machine learning.
>
> **4. Typos**
>
> Thank you for your careful reading! We have fixed that in the revision.

---

> > ### Comment · Reviewer_ag1C · 2021-08-31
> > **Update**
> >
> > I would like to thank the authors for their detailed response to my comments and the additional clarifications and experiments in their general response. Hopefully, the authors can revise their work accordingly based on all the comments of the reviewers.

---

### Official Review · Reviewer_6uVc · 2021-07-18

**Rating:** 7
**Confidence:** 3

**Summary:**

In this work the authors propose a gradient-based method for identifying the Pareto Front of a Multi-Objective Optimization problem.   More specifically, they propose two methods, one based on Stein Variational Gradient Descent, and another based on overdamped Langevin dynamics to evolve forward a set of particles to the (local) pareto front.

The approach is based on the multiple gradient descent (MGD) formulation where the next particle descent direction is chosen to maximise the slowest decreasing rate amongst the objectives.  Assuming particles are advected along an unknown vector field $\phi^*$ the authors introduce a variational formulation which selects the $\phi$ according to MGD objective while encouraging diversity through the usual entropy regularisation.   This gives rise to SVGD-like dynamics.   An analogous formulation for Langevin Dynamics is also proposed.

The authors provide some theoretical guarantees for both in terms of concentration of the dynamics to the Pareto front.  Results are demonstrated for a number of numerical experiments.


**Ethical Concerns:**

I can see no ethical concerns

**Limitations And Societal Impact:**

The authors have adequately addressed these

**Main Review:**

The proposed method is an interesting contribution provides an interesting and novel application of SVGD dynamics.   The fact that the resulting dynamics are not driven by a gradient adds interesting challenges, which the authors recognize.  Both the paper and the supplementary information was clearly written, and the proofs relatively straightforward to follow.  However,  I do feel that there are some issues which needed to be addressed more carefully.

* For example: in the conclusion the authors state: "Theoretically, analyzing our methods are challenging due to the lack of explicit form of the invariant distributions".   This presupposes the assumption that the dynamics do concentrate onto an invariant measure -- and if so -- is it unique.    Theorems 3.1, 3.2 and 3.3 pre-suppose the existence of $\rho^*$ and implicitly it is assumed that it is unique.  The authors should address why they believe this is true a bit more carefully.   In the case of Theorem 3.3 this should be discharged using relatively standard results, and I would imagine it to be the case in the other results.

* Theorem 3.2 doesn't quite characterise the concentration to the Pareto front as is claimed in the text, but rather it provides a bound for the weighted kernel mean embedding of $g^*$.   This is understandable from the form of the proof, however, it does complicate affairs as there is strong parameter dependence $(m_d, \sigma^2, etc)$ on the LHS of both inequalities.   Would it be possible to extract bounds, asymptotics, etc specifically for $g^*$?   If not, then the authors should write the LHS explicitly, using the kernel trick etc, to demonstrate the explicit dependence on the constants.

* The first part of Theorem 3.3 is essentially the Helmholtz-decomposition for weighted measures.  Ideally, the authors should cite this well-known result if possible, looking in the references of [8] for example which makes use of this result.

* The authors should explicitly define the notation $f_{[\rho^*]}$ somewhere a bit more clearly, as it was not immediately obvious on a first read of the text.

* I find the justification of the second factor on page 3 to be a bit unclear.  As written the argument suggests we want to maximise $-\mathbb{E}_{x\sim \rho}[\nabla \cdot \phi(x)]$, not as stated.

Minor comments:

l17: power systems

l31: have been

l80: gradient

l125: it's a divergence operator not a diversity operator

l143: $\rho^*(x) \propto \exp(-f^*(x)/\alpha)$.

l158 The notation is completely unclear, and needs to be written far more carefully.

**Time Spent Reviewing:**

3

---

> ### Author Response · Authors · 2021-08-10
> **Authors' Response to Reviewer 6uVc**
>
> **1. Existence and uniqueness of the invariant distribution?**
>
> Thank you for this very interesting question. Note that Langevin algorithm is actually an SDE, so the existence and uniqueness of $\rho^*$ has been investigated by [1]. SVGD density flow is more complicated due to the kernel embedding.
> But if we can show the density flow is tight, then  invariant distributions may exist due to argument likes the Krylov-Bogoliuv Theorem. However, additional work is needed to show that the invariant measure is unique. We have added these remarks after our theorems.
>
> [1] Mattingly J C, Stuart A M, Higham D J. Ergodicity for SDEs and approximations: locally Lipschitz vector fields and degenerate noise[J]. Stochastic processes and their applications, 2002, 101(2): 185-232.
>
> **2. Regarding Theorem 3.2 and Theorem 3.3**
>
>  Thank you for this very interesting point regarding Theorem 3.2.
> We have spelled out LHS of the inequalities in Theorem 3.2 explicitly. The revised result reads:
> \begin{align*}
> \| g^*_{[\rho^*]}\|^2_{H}&=\int \rho^*(x) k(x,y) \rho^*(y) g^*(x)^\top g^*(y)dxdy\\
> &\leq \int \rho^*(x) k(x,y) \rho^*(y) \|g^*(x)\|^2dxdy\leq\frac1{2c_1^2}(L+\alpha/\sigma^2)^2m^2_d\sigma^2\| 1_{[\rho^*]}\|^2_{H}.
> \end{align*}
>
> In fact, if one assumes $w=\min_{x}\int \rho^*(y) k(x,y)>0$, then we can establish an upper bound for the $L_2$ norm:
> $$
> \int \|g^*(x)\|^2\rho^*(x)dx\leq \frac{1}{2c_1^2w}(L+\alpha/\sigma^2)^2m^2_d\sigma^2\| 1_{[\rho^*]}\|^2_{H}.
> $$
> But in our perspective, $w$ is usually difficult to lower bound, so we don't think it is a good idea to claim that we have an upper bound for the $L_2$ norm.
>
> **3. Other issues**
>
> 1) We agree that we should have cited  a reference to Helmholtz decomposition. We have added a remark after Theorem 3.3.
>
> 2) We also added a detailed and clear definition of  $f_{[\rho]}$ before Theorem 3.3:
> "we define the kernel embedding of any function $f$ with density $\rho$ as:
> $
> f_{[\rho]}(x)=\int k(x,y)f(y)\rho(y)dy.$"
>
> 3) You are right that we missed a negative sign in the second factor on page 3. Thanks for pointing it out.
>
> 4) Thanks a lot for pointing out the typos. We have fixed them.

---

> > ### Comment · Reviewer_6uVc · 2021-08-23
> > **Response to Authors**
> >
> > I thank the authors for their detailed replies to my comments.  Based on the other reviewers' comments and the authors' responses I shall keep my score as is.
> >
> > I encourage the authors to revise the submission based on the very helpful comments provided by the reviewers.

---

### Author Response · Authors · 2021-08-10
**Authors' General Comment**

We thank the reviewers for the valuable feedback. We have revised the draft based on your comments and will make further improvement in the final draft. In this rebuttal, we first provide two general responses to all the reviewers, then address your individual questions. Please let us know if you have any further questions.

---

### Author Response · Authors · 2021-08-10
**General Response #1: Comparison with other MOO algorithms**

Thanks for pointing out the additional references on MOO, which we have cited and drawn discussions in the revision. We should note that the methods in the reference are of very different styles from our method. They are designed to work best in different settings and have their own pros and cons. In fact  each weakness  can be simultaneously an advantage. For example, the very idea of our method is to avoid the need of specifying any reference vectors, because uniform distribution on linear reference vectors, and object-ratio vectors (such as the one in EPO)  do not necessarily yield uniform distributions on the Pareto front (see for example the results on EPO in Figure 1 of the main text), while our method tries to adaptively achieve diverse distribution using the repulsive force. Similarly, providing multiple solutions to cover the Pareto frontier is another key advantage of our method. Given that the Pareto front can be a space of large area and users may have very different preference, it is fundamentally impossible to cover the need of all users using a single solution.

 **Pareto hypernet methods :**  Several methods, such as [1, 2, 5],  proposed to learn a neural network, called Pareto hypernet, which can generate pareto-optimal solutions given a preference vector as input. In comparison, our methods directly generates a set of solutions (or particles) that covers the  Pareto front. The usage of neural network can be helpful in some cases,
but it also makes the algorithm  more complicated and less transparent, and the choice of neural network structure has heavy influence on the result. In comparison, our algorithm is much simpler, transparent and directly outputs the solutions that we want.  At minimum, we certainly do not believe that the proposal of Pareto hypernets eliminates the need/room of designing novel particle based methods.

To further investigates the performance of Pareto hypernet methods, during the rebuttal, we tested the COSMOS method proposed in [5], which is the state-of-the-art of Pareto hypernets, with their official implementation on ZDT problems. The results are shown in the following link: <https://drive.google.com/file/d/1AF9QgTerpZvVKCoGWXpTwByVlDcq-RAB/view?usp=sharing>.

We find that:

(1) It still need to assign preference vectors manually (even though it is not needed theoretically if the network could be trained to be perfect), and a uniform assignment may not cause a uniform Pareto front, as we show in Fig. 1(a) and Fig. 1(b). In comparison, MOO-SVGD can generate an uniform Pareto front as we show in the paper.

(2) The methods based on hypernets may be limited by the optimization of the function approximator. In Fig 1.(c), several points are not on the real Pareto front. In contrast, we show both theoretically and empirically that MOO-SVGD converges to the Pareto front.

(3) Choosing appropriate hypernet structure can be tricky. In Fig. 1(c) and Fig. 1(d), we use a two-layer neural network and a three-layer neural network with similar number of parameters ($4800$ vs. $5520$).
We train both networks to convergence and find that they yield very different results. Note that MOO-SVGD only maintains $30 \times 50=1500$ parameters to store the solutions in this setting.

We will add comparison with these methods in later versions.

**Bayesian Optimization Methods :** [3] is an example of Bayesian optimization approaches for multi-objective optimization. There  is a huge body of related literature, but they are more suitable for optimizing unknown or highly expensive functions whose gradient is not available. Generally, they are not suitable for modern large-scale optimization problems like neural networks for which speed and gradient information is critical. We discussed the relationship with them in L224-229, and provide comparisons on ZDT problems with several representative algorithms in Appendix D.1.

**Hypervolume Methods :** [4] is mathematically similar to Hypervolume indicator gradient ascent [6] (HIGA),
which we have already discussed in L252-256 and Appendix F. HIGA and [4] have a number of drawbacks compared with our method empirically: (1) the result of both of them depends on the choice of the reference point, which need to be specified by the user based on an estimation of function range beforehand; in particular, if a particle excesses the range of the reference point, it would receive no gradient. (2) We found that the particles can overlap on the Pareto front. In comparison, in this case MOO-SVGD will move overlapping particles apart with the repulsive force.

**References:**

[1] Navon, A., Shamsian, A., Chechik, G., & Fetaya, E. (2020). Learning the Pareto front with hypernetworks. ICLR 2021.

[2] Lin, X., Yang, Z., Zhang, Q., & Kwong, S. (2020). Controllable Pareto multi-task learning. arXiv preprint arXiv:2010.06313.

[3] Shah, A., & Ghahramani, Z. (2016, June). Pareto frontier learning with expensive correlated objectives. In International Conference on Machine Learning (pp. 1919-1927). PMLR.

[4] Deist, T. M., Grewal, M., Dankers, F. J., Alderliesten, T., & Bosman, P. A. (2021). Multi-Objective Learning to Predict Pareto Fronts Using Hypervolume Maximization. arXiv preprint arXiv:2102.04523.

[5] Ruchte, M., & Grabocka, J. (2021). Efficient Multi-Objective Optimization for Deep Learning. arXiv preprint arXiv:2103.13392.

[6] Wang H, Deutz A, Bäck T, et al. Hypervolume indicator gradient ascent multi-objective optimization[C]//International conference on evolutionary multi-criterion optimization. Springer, Cham, 2017: 654-669.

---

### Author Response · Authors · 2021-08-10
**General Response #2: Regarding MOO-Langevin Dynamics**

We thank every reviewer for their valuable comments and efforts! Here we address some general issues related to MOO-LD.

**The necessity of MOO-LD :** Although MOO-LD does not perform as well as MOO-SVGD in our results,
we thought it is still conceptually very interesting (and seem to be easier to analyze theoretically than SVGD).
In addition, given that there is a much bigger  literature on Langevin dynamics and MCMC than SVGD, we thought it could be valuable to showcase the potential of combining MOO and MCMC, which seems to be an intersection that has not been well explored in the current literature. We hope the discussion on MOO-LD can motivate future works that improve over our method by other experts from MCMC literature. To avoid the confusion, we will clearly acknowledge the weakness of our current MOO-LD algorithm and our purpose of including them despite weaker empirical performance than MOO-SVGD.

**What causes the sub-optimal empirical performance of MOO-LD?**

We assume there are two reasons for its empirically worse performance:
(1) it requires longer burn-in time to effectively draw sample, and (2) the objectives in the experiments are non-convex.
We run additional experiments to validate our conjectures. Results are shown in <https://drive.google.com/file/d/1IMwNHqLO7qAke-bcCynNtUWThNG50WDf/view?usp=sharing>

For (1), we run MOO-LD for 5 million iterations on ZDT1. We observe that the algorithm is slowly making progress with more iterations. See Fig. 2(a).

For (2), in Appendix A, we show that MOO-LD works on par with MOO-SVGD on a 1d toy example with convex objectives $f_1(x) = x^2$ and $f_2(x) = (1-x)^2$. We generalize the problem to $d$ dimensions, with $f_1(x) = \frac{1}{d}\sum_{i=1}^d x_i^2, f_2(x) = \frac{1}{d}\sum_{i=1}^d (1-x_i)^2$. The results are shown in Fig. 2(b) and Fig. 2(c). We observe that MOO-LD still provide comparable Pareto front as MOO-SVGD even on 100-dimensional MOO problem. This verifies our theory which claims that MOO-LD converges to the true Pareto front for convex objectives, and also suggests the potential of MOO-LD. We leave the improvement of MOO-LD to high-dimensional non-convex problems to future works.

---

### Decision · Program_Chairs · 2021-09-27

**Decision:**

Accept (Spotlight)

**Comment:**

This paper proposes and studies a novel algorithm, inspired by Stein variational gradient descent, whose aim is to find distinct elements on the Pareto front of a multi-objective optimisation problem.  Theoretical and empirical results support the proposed method.  All reviewers agreed that the paper would make an excellent contribution to NeurIPS, but I suggest the author(s) carefully consider the comments of Reviewer 6uVc, who explains that there are elements of the presentation of theoretical results that ought to be clarified in the manuscript.